# Targeting dual-specificity tyrosine phosphorylation-regulated kinase 2 with a highly selective inhibitor for the treatment of prostate cancer

Kai Yuan[1,2,7], Zhaoxing Li [1,3,7], Wenbin Kuang[1,2,7], Xiao Wang[1,2,7], Minghui Ji[1,2], Weijiao Chen[1,2], Jiayu Ding[1,2], Jiaxing Li[1,2], Wenjian Min[1,2], Chengliang Sun[1,2], Xiuquan Ye[1,3], Meiling Lu[1,4], Liping Wang[1,2], Haixia Ge[5], Yuzhang Jiang [6✉], Haiping Hao[1,3✉], Yibei Xiao[1,3✉] & Peng Yang [1,2✉]

Prostate cancer (PCa) is one of the most prevalent cancers in men worldwide, and hormonal therapy plays a key role in the treatment of PCa. However, the drug resistance of hormonal therapy makes it urgent and necessary to identify novel targets for PCa treatment. Herein, dual-specificity tyrosine phosphorylation-regulated kinase 2 (DYRK2) is found and confirmed to be highly expressed in the PCa tissues and cells, and knock-down of DYRK2 remarkably reduces PCa burden in vitro and in vivo. On the base of DYRK2 acting as a promising target, we further discover a highly selective DYRK2 inhibitor YK-2-69, which specifically interacts with Lys-231 and Lys-234 in the co-crystal structure. Especially, YK-2-69 exhibits more potent anti-PCa efficacy than the first-line drug enzalutamide in vivo. Meanwhile, YK-2-69 displays favorable safety properties with a maximal tolerable dose of more than 10,000 mg/kg and pharmacokinetic profiles with 56% bioavailability. In summary, we identify DYRK2 as a potential drug target and verify its critical roles in PCa. Meanwhile, we discover a highly selective DYRK2 inhibitor with favorable druggability for the treatment of PCa.

[1] State Key Laboratory of Natural Medicines and Jiangsu Key Laboratory of Drug Design and Optimization, China Pharmaceutical University, 210009 Nanjing, China. [2] Department of Medicinal Chemistry, School of Pharmacy, China Pharmaceutical University, 211198 Nanjing, China. [3] Department of Pharmacology, School of Pharmacy, China Pharmaceutical University, 211198 Nanjing, China. [4] School of Life Science and Technology, China Pharmaceutical University, 211198 Nanjing, China. [5] School of Life Sciences, Huzhou University, 313000 Huzhou, China. [6] Department of Laboratory, Huai'an First People's Hospital, Nanjing Medical University, 223300 Huai'an, Jiangsu, China. [7] These authors contributed equally: Kai Yuan, Zhaoxing Li, Wenbin Kuang, Xiao Wang. ✉email: jyz8848@163.com; hhp_770505@hotmail.com; yibei.xiao@cpu.edu.cn; pengyang@cpu.edu.cn

Prostate cancer (PCa) is one of the most common cancers in men with an estimated incidence of 268,490 new cases in the U.S. alone, accounting for 27% of all new cancer cases in U.S. males in 2022[1,2]. The mortality rate of PCa ranks second in U.S. male cancer and is expected to reach 11% of all male cancer deaths in 2022[1,2]. The majority of PCa primarily relies on androgens for survival and growth, and PCa treatment mainly focused on reducing hormone levels[3–5]. Currently, hormonal therapy, including the antiandrogens[6,7], gonadotropin-releasing hormone (GnRH) agonists and antagonists[8,9], androgen biosynthesis inhibitor[10,11], and androgen receptor inhibitor[12,13], has been developed. However, hormonal therapy can only delay PCa progression but is not a curative method for PCa treatment[14,15]. Ultimately, drug resistance of hormonal therapy is easily generated and most PCa patients developed to a metastatic, hormone-resistant state[16,17]. Therefore, despite the leading role of androgen in the PCa, it is necessary to identify novel regulators of PCa and target them for PCa treatment.

Dual-specificity tyrosine phosphorylation-regulated kinases (DYRKs) are evolutionarily conserved enzymes, and the most remarkable characteristic property of them is to display both tyrosine and serine/threonine kinase activities. DYRKs can be divided into class I (DYRK1A and DYRK1B) and class II (DYRK2, DYRK3 and DYRK4) in mammals. Class I and class II DYRKs are mainly localized in the nucleus and cytoplasm, respectively. Among class II DYRKs, DYRK2 is most deeply studied, which plays important while controversial roles in human cancers[18,19]. Previously, DYRK2 was primarily considered as a cancer suppressor, which can promote phosphorylation of P53 to induce apoptosis[20,21], facilitate degradation of c-JUN and c-MYC to inhibit the transition of the cell cycle from G1 to S phase[22,23], and accelerate the degradation of snail to suppress epithelial-to-mesenchymal transition (EMT) and cell migration and so on[24,25]. Recently, researches on the DYRK2 have gradually revealed its distinct role as an oncogene[26–28]. In multiple myeloma (MM) and triple-negative breast cancer (TNBC), DYRK2 phosphorylates Rpt3-Thr25 of the 26S proteasome to activate the 26S proteasome, and then promotes the transition of the cell cycle from G1 to S phase[29,30]. The inhibition of DYRK2 impedes 26S proteasome activity and suppresses the cell cycle progression to inhibit cell proliferation[31,32]. Therefore, DYRK2 is a potential target for the treatment of MM and TNBC. These researches revealed the critical and multifarious roles of DYRK2 in different cancers. However, the regulation mechanism of DYRK2 in PCa is still unclear and has not been reported. Moreover, the reported small-molecule DYRK2 inhibitors lack selectivity and exhibit poor druggability. Thus, it is urgent to reveal the function of DYRK2 in PCa and develop DYRK2 inhibitors with better potency and higher selectivity to treat PCa and other cancers.

In our work, DYRK2 was identified as a potential target for PCa treatment. High expression of DYRK2 was detected and confirmed in both PCa patients and cell lines. Knock-down of DYRK2 in PCa cells suppressed cell proliferation and metastasis, promoted apoptosis, and caused a G1 arrest of the cell cycle. Furthermore, knock-down of DYRK2 significantly inhibited tumor growth of PCa in a xenograft model. Through virtual screening and structural optimization, we developed a unique DYRK2 inhibitor YK-2-69 with high selectivity over 370 kinases, and the detailed interactions between YK-2-69 and DYRK2 were further demonstrated by their co-crystal structure. Moreover, YK-2-69 displayed acceptable safety properties, favorable pharmacokinetic profiles, and stronger suppression of PCa progression than the first-line PCa drug enzalutamide in vivo. Therefore, DYRK2 was a biomarker in PCa diagnosis and a potential target to develop anti-PCa drugs. The DYRK2 inhibitor YK-2-69 with high selectivity and favorable druggability provided a potential candidate for the treatment of PCa.

## Results

**Highly expressed DYRK2 was a potential target in PCa.** To investigate the role of DYRK2 in PCa, we first mined The Cancer Genome Atlas (TCGA) to analyze the expression of *DYRK2* in normal and PCa patients. We found the higher expression of *DYRK2* in PCa patients when compared with normal controls (Fig. 1a). Also, different from PCa patients with intermediate risk and below the age of 65, the expression of *DYRK2* was significantly higher in high risk and above the age of 65 PCa patients, respectively (Fig. 1b, c). Importantly, the relapse-free survival (RFS) in patients with low expression of *DYRK2* was remarkably better than those with high expression of *DYRK2* (Fig. 1d). Therefore, DYRK2 was the potential target for PCa treatment based on analysis of TCGA. Meanwhile, other DYRK family members, DYRK1A, DYRK1B, DYRK3, and DYRK4 were not great candidate targets for anti-PCa drugs based on analysis of TCGA (Supplementary Fig. 1). Furthermore, the *DYRK2* mRNA levels of malignancy PCa tissues were higher than adjacent normal prostate tissues in PCa patients (Fig. 1e). Immunohistochemistry of DYRK2 in these patient-derived PCa tumors also demonstrated that the expression of DYRK2 in tumor tissues was much higher than in normal tissues (Fig. 1f). Meanwhile, the protein levels of DYRK2 in PCa cells were further determined by western blotting analysis, and the results showed the higher expression of DYRK2 in DU145, PC-3, and 22Rv1 cells compared with prostate RWPE-1 cells (Fig. 1g). All these results indicated that DYRK2 is highly expressed in PCa, which could be a potential target to develop anti-PCa drugs.

**Knock-down of DYRK2 remarkably reduced PCa burden.** To further study the role of DYRK2 in PCa, we knocked down DYRK2 in DU145 and 22Rv1 cells using shRNAs (Fig. 2a, b and Supplementary Fig. 2a). DYRK2 depletion significantly suppressed the cell proliferation (Fig. 2c, d and Supplementary Fig. 2b), migration (Supplementary Fig. 2c) and invasion (Supplementary Fig. 2d) in PCa cells. In addition, knock-down of DYRK2 caused a G0/G1 arrest of the cell cycle (Fig. 2e, f) and induced apoptosis (Fig. 2g, h). The cell cycle-related proteins, including p-RB, CDK4, and CDK6, which promoted the cell cycle progression, were down-regulated in the DU145 shDYRK2 cell. In contrast, cyclin-dependent kinase inhibitors (CKIs) P21 and P27, which inhibited the cell cycle, were up-regulated (Fig. 2i). The cell apoptosis-related proteins P53 and cleaved PARP were up-regulated, and XIAP was down-regulated. The increased expression of E-cadherin was also detected, which demonstrated metastasis was suppressed (Fig. 2i). Furthermore, to determine the effects of DYRK2 knock-down on PCa growth in vivo, we subcutaneously implanted the DU145 shNC and shDYRK2 cells into the nude mice. The tumor growth was significantly inhibited (Fig. 2j) while the body weight of mice increased normally when compared with the shNC group (Supplementary Fig. 2e). H&E staining and Ki-67 immunohistochemical analysis of tumor tissues indicated that knock-down of DYRK2 exhibited potent efficacy of killing tumor cells and inhibiting PCa cell proliferation (Supplementary Fig. 2f, g). The WB analysis of the tumor tissues demonstrated that p-RB, CDK4, CDK6, and XIAP were down-regulated, while P27, P53, and cleaved PARP were up-regulated in vivo (Fig. 2k). Furthermore, we also inoculated subcutaneously 22Rv1 shNC and shDYRK2 cells into mice. Similar as the results in DU145 shDYRK2 studies, the body weight of mice increased normally but no visible tumors were detected in the 22Rv1 shDYRK2 group (Supplementary Fig. 2h, i). In summary, the down-regulation of DYRK2 remarkably reduced PCa tumor burden in vitro and in vivo, suggesting that DYRK2 played a critical function in regulating PCa and is a potential therapeutic target for the treatment of PCa.

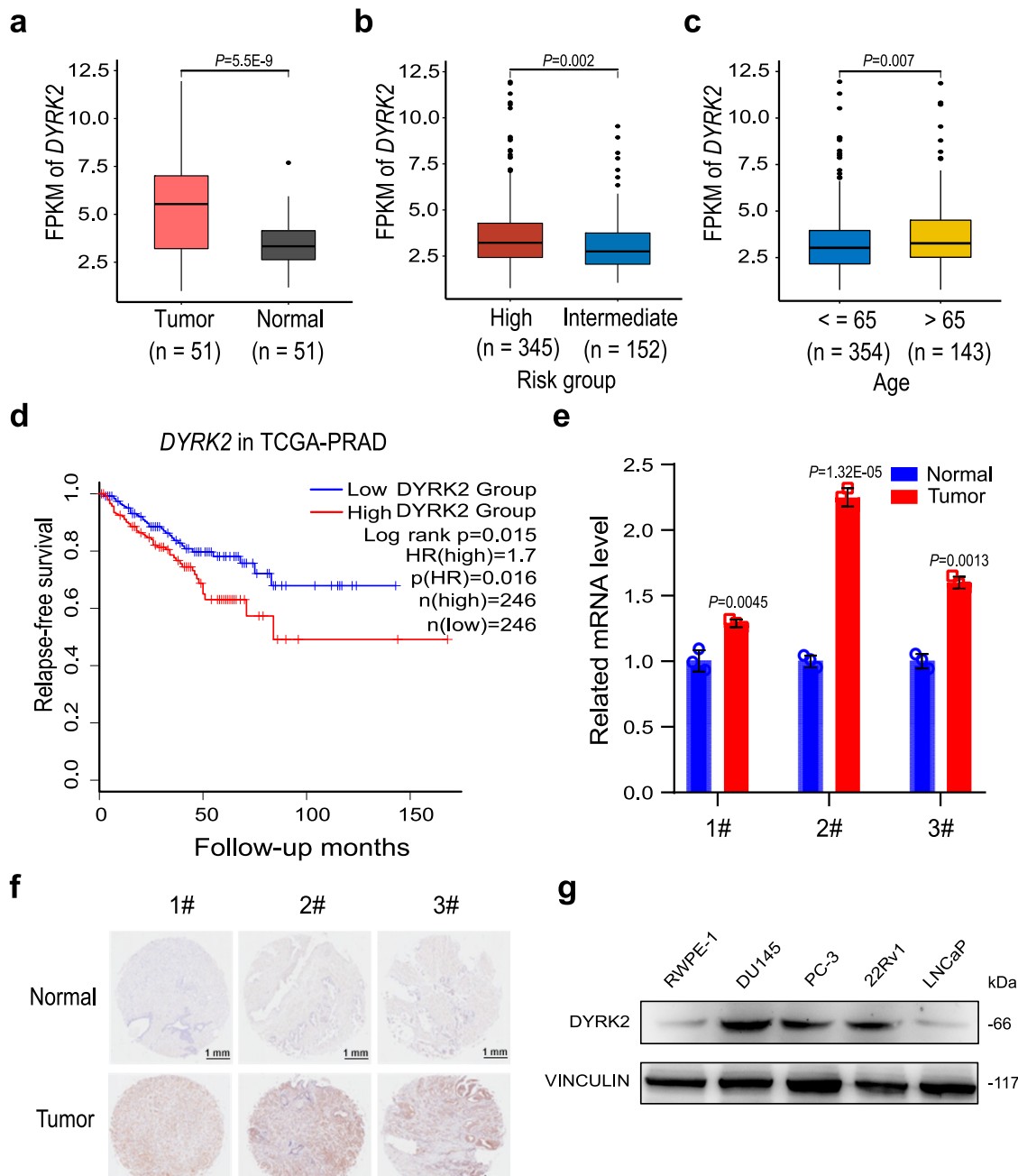

**Fig. 1 DYRK2 is highly expressed in PCa. a–c** Comparison of *DYRK2* expression between tumor (red, $n = 51$) and normal (gray, $n = 51$) tissues (**a**), high (red, $n = 345$) and intermediate (blue, $n = 152$) risk PCa patients (**b**), age ≤65 (blue, $n = 354$) or > 65 (yellow, $n = 143$) PCa patients (**c**) in TCGA database. FRKM: fragment per killo million. The whiskers of boxplot represent the quantile percentile, from bottom to top are minima, 25%, median, 75%, and maxima respectively. Two-tailed Student's *t* test was applied without adjustment for multiple comparisons (false discovery rate, FDR). **d** Kaplan–Meier survival plot of high (red line, $n = 246$) and low (blue line, $n = 246$) *DYRK2* expression PCa patients. Log-rank test, $P = 0.015$. **e, f** Analysis of DYRK2 expression in three PCa patients. *DYRK2* mRNA level (**e**) and immunohistochemical analysis of DYRK2 expression (**f**) in tumor and normal tissues. Unpaired two-tailed Student's *t* test. Error bar, mean ± SD, $n = 3$. **g** DYRK2 protein levels in different PCa cell lines. Normal prostate epithelial cell RWPE-1 was used as the control. Source data are provided as a Source Data file.

**YK-2-69 was discovered as a highly selective DYRK2 inhibitor.** Considering the high expression level and critical regulation roles of DYRK2 in PCa, we took DYRK2 as a potential drug target and conducted a structure-based virtual screening of the Specs database and an in-house library to identify DYRK2 inhibitors (Fig. 3a, see Supplementary Methods for details). Compounds with poor drug-like properties were first filtered[33–36], then the remained 195,483 compounds were subjected to structure-based virtual screening (DYRK2 PDB ID: 6K0J) via the Libdock and CDOCKER protocol of Discovery Studio 2020 (DS2020, Accelrys, CA, USA)[32,37,38]. 2,724 ligands were reserved and further clustered into 100 clusters, then 15 compounds were selected for DYRK2 inhibitory activity evaluation (Supplementary Fig. 3)[39]. Among these 15 compounds, compound **12** was identified as a top hit, which displayed potent inhibition on DYRK2 with a half maximal inhibitory concentration (IC$_{50}$) value of 263 nM (Supplementary Table 1). Through systematic optimization (see Methods for details), multiple series of derivatives **16-27**

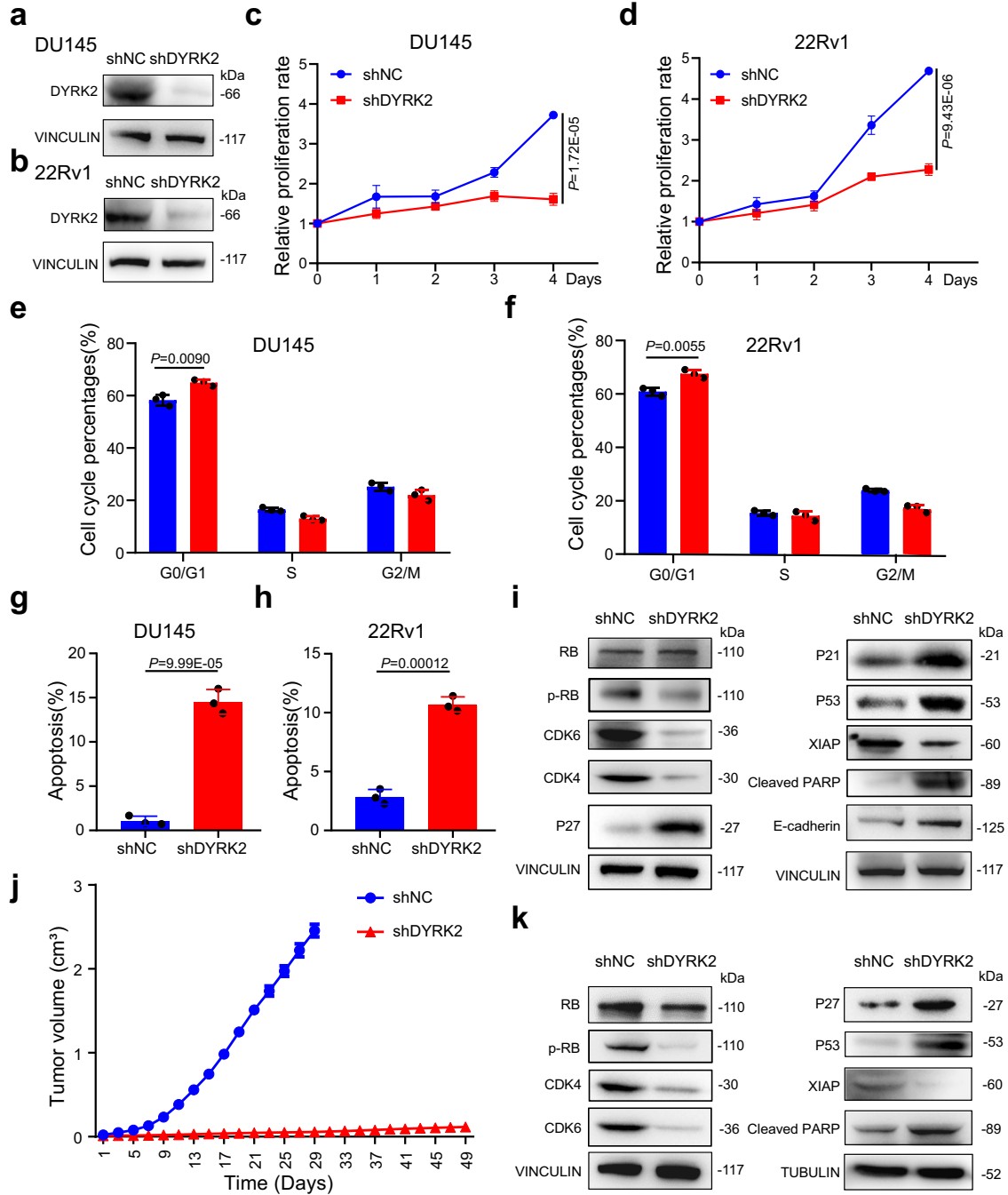

**Fig. 2 Knock-down of DYRK2 inhibited PCa in vitro and in vivo. a, b** Protein level of DYRK2 in DU145 shNC/shDYRK2 (**a**) and 22Rv1 shNC/shDYRK2 (**b**) cells. **c, d** Cell viability of DU145 shNC/shDYRK2 cells (**c**) and 22Rv1 shNC/shDYRK2 cells (**d**) during a 5-day course. Unpaired two-tailed Student's $t$ test. Error bar, mean ± SD, $n = 3$. **e, f** Cell cycle phase distribution of DU145 shNC/shDYRK2 cells (**e**) and 22Rv1 shNC/shDYRK2 cells (**f**) determined by flow cytometry. Unpaired two-tailed Student's $t$ test. Error bar, mean ± SD, $n = 3$. **g, h** Apoptosis of DU145 shNC/shDYRK2 cells (**g**) and 22Rv1 shNC/shDYRK2 cells (**h**) determined by flow cytometry. Unpaired two-tailed Student's $t$ test. Error bar, mean ± SD, $n = 3$. **i** Western blotting analysis of indicated proteins in DU145 shNC and shDYRK2 cells. **j, k** BALB/c nude mice were implanted subcutaneously with DU145 shNC ($n = 6$) and shDYRK2 ($n = 10$) cells. Tumor volume of mice (**j**) was measured every two days. Error bar, mean ± SD. The shNC group was euthanatized at 29th day and shDYRK2 group was euthanatized at 49th day. Tumor tissues of mice treated with DU145 shDYRK2 and shNC cells were taken out, then the total proteins in the tumor were extracted and subjected to the western blotting analysis of indicated proteins (**k**). Source data are provided as a Source Data file.

(Supplementary Fig. 4) were synthesized. Among them, compound **26** (re-named as YK-2-69) exhibited the most potent DYRK2 inhibitory activity with an IC$_{50}$ value of 9 nM (Fig. 3b). In particular, YK-2-69 showed selectivity to the DYRK subfamily with 60-fold selectivity over DYRK1B and more than 100-fold selectivity over DYRK1A, DYRK3, and DYRK4 (Fig. 3c). To

further estimate the kinase selectivity of YK-2-69, its inhibitory activities against 370 kinases were tested. Besides DYRK2 and CDK4/6[40], YK-2-69 exhibited selectivity over 360+ kinases (Fig. 3d and Supplementary Fig. 5). In addition, the $K$d values of lead compound **12** and YK-2-69 with DYRK2 were 4.21 μM and 92 nM, respectively (Supplementary Fig. 6a, b). Taken together,

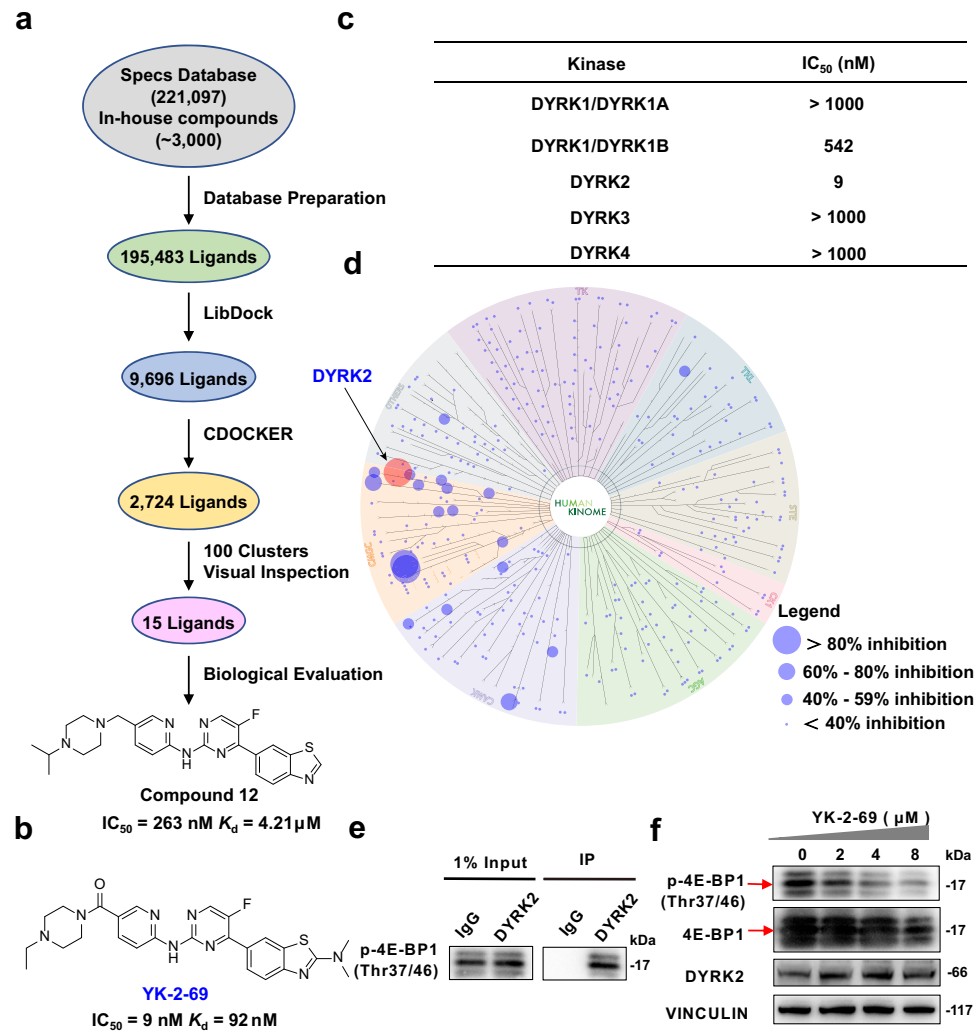

**Fig. 3 Discovery of the highly selective DYRK2 inhibitor YK-2-69. a** Flowchart of virtual screening to discover the hit DYRK2 inhibitor compound **12**. **b** Chemical structure and activities of YK-2-69. **c** Inhibitory activity of YK-2-69 against DYRK1A, 1B, 2, 3, and 4. **d** Kinase selectivity experiment of YK-2-69 (1 µM) was carried out at Reaction Biology Corporation (https://www.reactionbiology.com/). The dot size indicates the inhibitory rate. A dot represents a type of kinase and the red dot represents DYRK2. **e** Validation of interaction of p-4E-BP1 (Thr37/46) with DYRK2 by co-immunoprecipitation assay. 1% volume of cell lysate was used as input. **f** Western blotting analysis of indicated proteins after treatment with DMSO or YK-2-69 for 48 h. The special bands of 4E-BP1 and p-4E-BP1 were shown with arrows. Source data are provided as a Source Data file.

YK-2-69 was discovered as a DYRK2 ligand with potent inhibitory activity and high selectivity. The previous report reveals that 4E-binding protein 1 (4E-BP1) is a direct cellular target of DYRK2, and DYRK2 can directly phosphorylate 4E-BP1[41,42]. Meanwhile, the phosphorylation of 4E-BP1 contributes to cell proliferation and tumor growth[43–45]. Our results revealed that p-4E-BP1 (Thr37/46) could interact with DYRK2 by the immunoprecipitation assay (Fig. 3e). Furthermore, YK-2-69 inhibited the phosphorylation of 4E-BP1 in a dose-dependent manner (Fig. 3f). These results demonstrated that YK-2-69 selectively binds to DYRK2 and inhibits its kinase activity.

To explore the exact interaction of YK-2-69 with DYRK2 and elucidate the mechanism, we determined the co-crystal structure of YK-2-69 with DYRK2 at a high resolution of 2.5 Å (PDB ID: 7EJV, Fig. 4a and Supplementary Table 2). The co-crystal structure showed that YK-2-69 occupied the ATP-binding pocket of DYRK2, thereby preventing DYRK from exerting its enzymatic activity (Fig. 4b). The occupancy of the ATP-binding pocket is the same with all reported co-crystal structures[31,32,46–48]. The benzothiazole and pyrimidine rings were located deep into the ATP binding site. This orientation placed the pyrimidine ring

adjacent to the Lys-231. Lys-231 played a critical role in forming the hydrogen bond in the previous co-crystal structures (PDB ID: 3KVW, 4AZF, 6HDR)[46]. The pyrimidine ring and the linked secondary amine of YK-2-69 also interacted with Lys-231 in the formation of two hydrogen bonds. The tailed piperazine ring extended out and the conjoint carbonyl formed a hydrogen bond with the amino side chain of Asn-234 (Fig. 4c).

**YK-2-69 significantly inhibited growth and migration of PCa cells in vitro.** Once we confirmed YK-2-69 as a potent and selective DYRK2 inhibitor, we further investigated its effects on PCa cells. YK-2-69 showed potent inhibitory activity against the proliferation of DU145, PC-3, and 22Rv1 cells (Supplementary Fig. 7a–c). But for DU145 and 22Rv1 shDYRK2 cells, YK-2-69 exhibited almost no inhibitory activity on the proliferation even at 80 µM (Fig. 5a), which further confirmed the selective on-target activity of YK-2-69 to DYRK2. Meanwhile, YK-2-69 significantly inhibited the cell proliferation of DU145, PC-3, and 22Rv1 cells in a dose-dependent manner (Fig. 5b–e and Supplementary Fig. 7d, e), which was similar to knocking down DYRK2 in the PCa cells. Same as depletion of DYRK2 reduced EMT, YK-2-69 also remarkably inhibited the

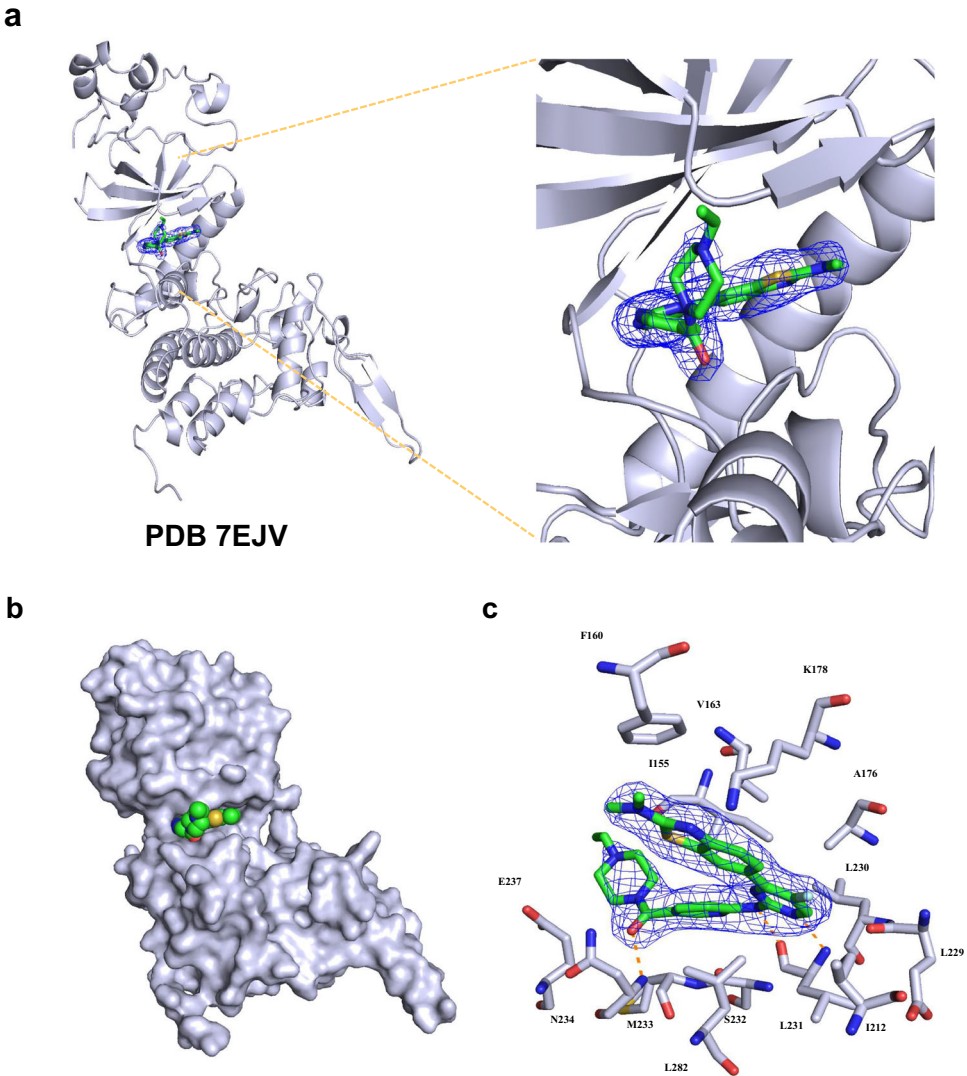

**Fig. 4 Co-crystal structure of YK-2-69 with DYRK2. a** The $F_O$-$F_C$ omitted map is contoured at 3.0 σ and shown as a blue mesh, which reveals the presence of YK-2-69. DYRK2 is shown as ribbons. PDB ID: 7EJV. **b** YK-2-69 occupies the ATP binding pocket of DYRK2. DYRK2 is shown as surface, and YK-2-69 is shown as spheres. **c** Detailed interactions between YK-2-69 and DYRK2 in the co-crystal structure.

migration and invasion of DU145, PC-3, and 22Rv1 cells in a dose-dependent manner (Fig. 5f–i and Supplementary Fig. 7f, g). All these results confirmed that YK-2-69 reduced cell proliferation and EMT through inhibiting DYRK2.

To investigate the possible role of YK-2-69 on the cell cycle and apoptosis, we conducted flow cytometry analysis. The treatment of DU145, PC-3, and 22Rv1 cells with YK-2-69 caused a significant increase of the apoptotic cell population (Fig. 5j, k and Supplementary Fig. 7h) and arrested cell cycle at the G0/G1 phase (Fig. 5l, m and Supplementary Fig. 7i) in a concentration-dependent manner. Furthermore, the DU145 cells treated with YK-2-69 decreased cell cycle-related protein levels of p-RB, CDK4 and CDK6 as well increased P21, increased apoptosis-related protein levels of P53 and cleaved PARP as well decreased XIAP, and also increased expression of E-cadherin (Supplementary Fig. 7j). In summary, knock-down of DYRK2 and small-molecule inhibitor YK-2-69 displayed the similar effects on PCa cells in vitro, which inhibited cell growth through G0/G1 arrest and apoptosis induction, and decreased the EMT activity.

**DYRK2 KD targeted similar signaling pathways to YK-2-69 treatment in the proliferation inhibition of PCa cells.** To

investigate which signaling pathways are responsible for the anti-prostate cancer function of DYRK2 inhibitors, we performed transcriptome-wide RNA-sequencing analysis of DYRK2 KD- and YK-2-69- treated human DU145 and 22Rv1 cells as well as control cells[49,50]. Many signaling pathways regulated by DYRK2 KD could also be regulated by YK-2-69 treatment, especially the vast majority of pathways (28 out of 34, 82.4%; 46 out of 48, 95.8%) inhibited by DYRK2 KD were also suppressed by YK-2-69 treatment (Fig. 6a and Supplementary Fig. 8a). By independent analysis of two different comparisons, we found that both DYRK2 KD and YK-2-69 treatment induced significant inhibition of MYC targets (Fig. 6b and Supplementary Fig. 8b), which may contribute to the inhibitory effects of DYRK2 KD and YK-2-69 treatment on cell cycle and proliferation. Moreover, we made shDYRK2 and YK-2-69 as a single group and re-analyzed the sequencing data between this group and its control group (shNC group and DMSO group). Consistently, we found that YK-2-69 treatment and DYRK2 KD inhibited cell cycle and proliferation related signaling pathways MYC target V1, MYC target V2, E2F targets (Fig. 6c). DYRK2 KD and YK-2-69 treatment significantly down-regulated genes enriched in MYC target V1, MYC target V2 and Mitotic SPINDLE (Fig. 6d). Then, we also found that

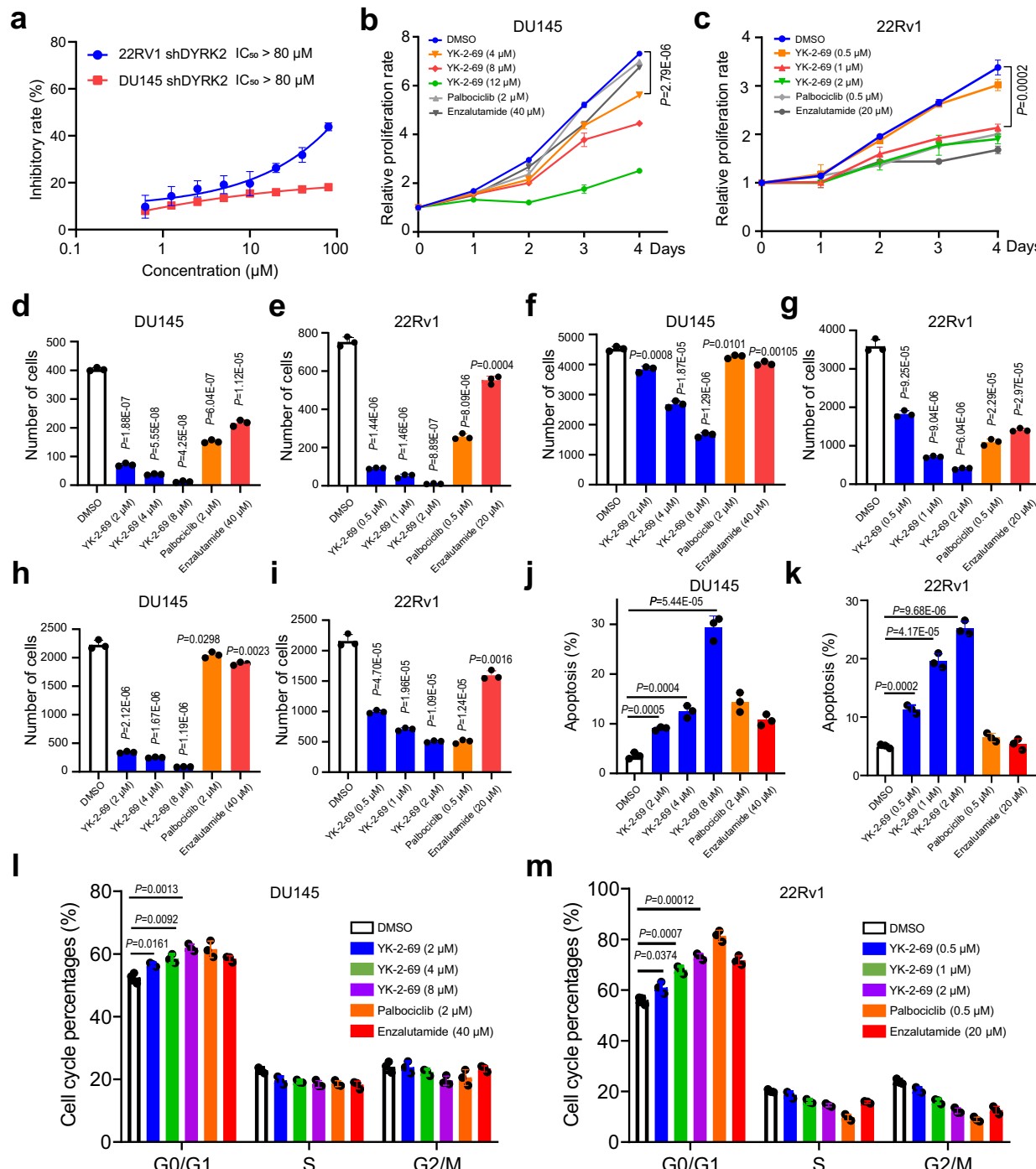

**Fig. 5 YK-2-69 inhibited cell growth and metastasis and induced apoptosis in PCa cells. a** Antiproliferative activity of YK-2-69 against DU145 and 22Rv1 shDYRK2 cells. Error bar, mean ± SD, $n = 3$. **b**, **c** Effects of YK-2-69 (4, 8, 12 μM or 0.5, 1, 2 μM), palbociclib (2 or 0.5 μM), or enzalutamide (40 or 20 μM) treatment on the DU145 (**b**) and 22Rv1 (**c**) cells viability during a 5-day course. Unpaired two-tailed Student's *t* test. Error bar, mean ± SD, $n = 3$. **d**, **e** Quantification of DU145 (**d**) and 22Rv1 (**e**) cells colony numbers. Before being plated on the 24-well plate for colony formation, DU145 and 22Rv1 cells were treated with DMSO, YK-2-69 (2, 4, 8, or 0.5, 1, 2 μM), palbociclib (2 or 0.5 μM), or enzalutamide (40 or 20 μM) for 48 h. Unpaired two-tailed Student's *t* test. Error bar, mean ± SD, $n = 3$. **f**, **g** Quantification of migration ability of DU145 (**f**) and 22Rv1 (**g**) cells after treatment with DMSO, YK-2-69 (2, 4, 8 μM or 0.5, 1, 2 μM), palbociclib (2 or 0.5 μM), or enzalutamide (40 or 20 μM) for 48 h. Unpaired two-tailed Student's *t* test. Error bar, mean ± SD, $n = 3$. **h**, **i** Quantification of invasion ability of DU145 (**h**) and 22Rv1(**i**) cells after treatment with DMSO, YK-2-69 (2, 4, 8 or 0.5, 1, 2 μM), palbociclib (2 or 0.5 μM), or enzalutamide (40 or 20 μM) for 48 h. Unpaired two-tailed Student's *t* test. Error bar, mean ± SD, $n = 3$. **j**, **k** Apoptosis of DU145 (**j**) and 22Rv1 (**k**) cells after treatment with DMSO, YK-2-69 (2, 4, 8 or 0.5, 1, 2 μM), palbociclib (2 or 0.5 μM), or enzalutamide (40 or 20 μM) for 48 h determined by flow cytometry. Unpaired two-tailed Student's *t* test. Error bar, mean ± SD, $n = 3$. **l**, **m** Cell cycle phase distribution of DU145 (**l**) and 22Rv1 (**m**) cells after treatment with DMSO, YK-2-69 (2, 4, 8 or 0.5, 1, 2 μM), palbociclib (2 or 0.5 μM), or enzalutamide (40 or 20 μM) for 48 h determined by flow cytometry. Unpaired two-tailed Student's *t* test. Error bar, mean ± SD, $n = 3$. Source data are provided as a Source Data file.

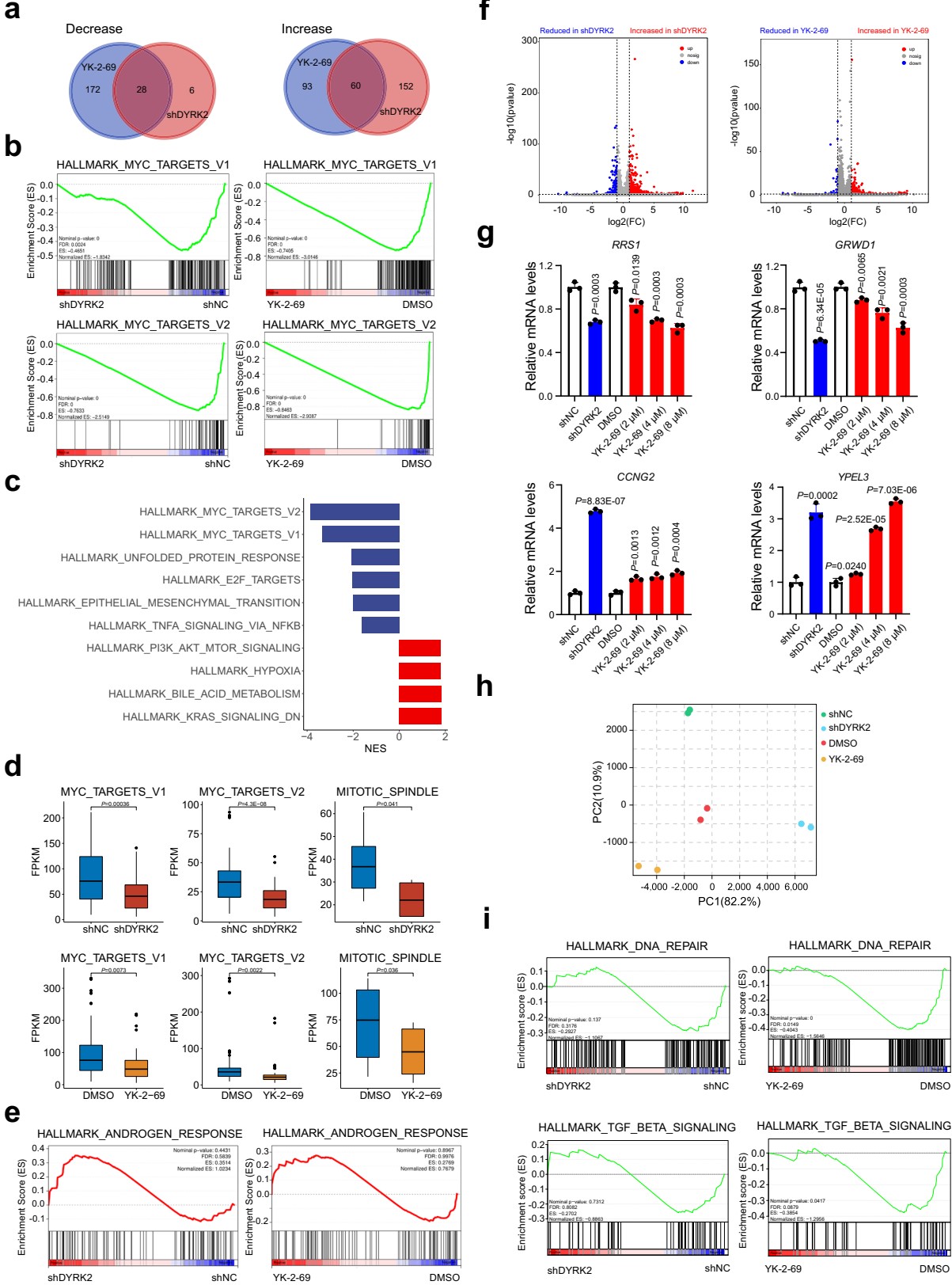

DYRK2 KD and YK-2-69 treatment did not affect the ANDROGEN RESPONSE signaling pathway, while enzalutamide significantly inhibited the ANDROGEN RESPONSE signaling pathway (Fig. 6e and Supplementary Fig. 8c). In summary, these data suggested that DYRK2 KD and YK-2-69 treatment played important and similar roles in cell proliferation inhibition.

Through the analysis of differentially expressed genes (DEG) between shDYRK2 and YK-2-69-treated samples in transcriptome-wide RNA-sequencing (Fig. 6f and Supplementary Dataset 1) and the further experiment confirmation, we found significant expression changes in human regulator ribosome synthesis 1 (RRS1), glutamate-rich WD40 repeat containing 1

**Fig. 6 Transcriptome-wide RNA sequencing assays in PCa DU145 cells. a** Transcriptome strategy of RNA-sequencing conducted on DU145 cells exposed to YK-2-69 (3 μM) for 48 h. The shNC, shDYRK2, DMSO, and YK-2-69 groups all contain two biological replicates. Venn diagram of upregulated and downregulated signaling pathways in DYRK2 KD- and YK-2-69-treated DU145 cells. The number of genes in every signaling pathway is >50. Normalized enrichment score (NES) >1 or <−1; $P < 0.05$; FDR < 0.25. **b** The signaling pathways enriched in different groups obtained through Gene Set Enrichment Analysis (GSEA). **c** The core-enriched decreased (blue) and increased (red) signaling pathways in shDYRK2 and YK-2-69 treatment groups when compared with shNC and DMSO groups, respectively. The signaling pathways with $P < 0.05$ are presented. **d** The relative abundance of genes involved in MYC target V1, MYC target V2 and Mitotic SPINDLE in DYRK2 KD- and YK-2-69-treated DU145 cells. $n = 2$. The whiskers of boxplot represent the quantile percentile, from bottom to top are minima, 25%, median, 75%, and maxima respectively. Two-tailed Student's $t$ test was applied without adjustment for multiple comparisons (FDR). **e** GSEA was used to analyze the effects of DYRK2 KD or YK-2-69 treatment on the ANDROGEN RESPONSE signaling pathway in DU145 cells. **f** Volcano plot of significantly affected genes (absolute fold change > 2, $P < 0.05$) in DU-145 shDYRK2 group relative to shNC group and YK-2-69 group relative to DMSO group. The negative binomial distribution test of DESeq2 software was used. **g** Effects of DYRK2 KD or YK-2-69 treatment on the *RRS1*, *GRWD1*, *CCNG2*, and *YPEL3* mRNA levels in DU145 cells. Unpaired two-tailed Student's $t$ test. Error bar, mean ± SD, $n = 3$. **h** The principal component analysis was used to identify transcriptome differences between two samples. **i** The different signaling pathways enriched in DYRK2 KD and YK-2-69 treatment groups obtained through GSEA. Source data are provided as a Source Data file.

**Table 1 Pharmacokinetic parameters of compound YK-2-69 in SD rats[a].**

| Compd. | Admin. | $C_{max}$ (ng/mL) | $AUC_{0-\infty}$ (h·ng/mL) | $MRT_{0-\infty}$ (h) | $T_{max}$ (min) | $t_{1/2}$ (h) | CL (mL/h/kg) | F (%) |
|---|---|---|---|---|---|---|---|---|
| YK-2-69 | IV | 974 | 1503 | 3.5 | 2 | 3 | 669 | – |
| | PO | 674 | 8384 | 8.9 | 240 | 5 | 1198 | 56 |

Dose: p.o. at 10 mg/kg. Dose: i.v. at 1 mg/kg. Source data are provided as a Source Data file.
$C_{max}$ maximum concentration, *AUC* area under the plasma concentration-time curve, *MRT* mean residence time, $t_{1/2}$ half-life, *CL* clearance, *F* oral bioavailability.
[a]Values are the average of three runs.

(*GRWD1*), cyclin G2 (*CCNG2*), and Yippee-like-3 (*YPEL3*). RRS1, an essential nuclear protein involved in ribosome biogenesis, is overexpressed in some human cancers[51], and downregulation of RRS1 causes a G1 arrest of cell cycle[52]. GRWD1, a negative transcriptional regulator of P53, plays an oncogenic activity in human cancers[53,54]. Meanwhile, previous studies have shown that overexpression of CCNG2 can induce apoptosis and inhibit cell proliferation[55,56]. YPEL3, a P53-regulated gene, has been reported to display growth suppressive and EMT inhibitory activity[57,58]. The experimental results of qRT-PCR demonstrated that oncogenes *RRS1* and *GRWD1* were downregulated and tumor suppressors *CCNG2* and *YPEL3* were upregulated regardless of knocking down DYRK2 or YK-2-69 treatment (Fig. 6g and Supplementary Fig. 8d). These results suggested that *RRS1*, *GRWD1*, *CCNG2*, and *YPEL3* may play important roles in the DYRK2 regulation mechanism and YK-2-69 treatment to PCa cells.

The principal component analysis indicated that DYRK2 KD and YK-2-69 treatment also induced some different transcriptomic changes (Fig. 6h). Therefore, the further Gene Set Enrichment Analysis (GSEA) between shDYRK2 and YK-2-69 was conducted, which demonstrated that YK-2-69 treatment induced inhibition of DNA repair and TGF β signaling pathways, while DYRK2 KD had no remarkable effects on these two signaling pathways (Fig. 6i). Inhibition of DNA repair is a successful therapeutic strategy for cancers with several approved drugs in the market[59–61]. TGF-β also plays a critical role as a tumor promoter in late-stage cancer[62,63], and a number of drugs for inhibiting TGF β signaling pathway have been developed and evaluated in clinical trials[64]. Therefore, YK-2-69 may regulate some different signaling pathways to generate potent antitumor activity when compared with DYRK2 KD.

**YK-2-69 displayed favorable safety properties and pharmacokinetic profiles.** To evaluate the toxic effects of DYRK2 inhibitor YK-2-69 in vivo, the ICR mice ($n = 10$/group) were orally administrated YK-2-69 in the single dose of 2500 mg/kg,

5000 mg/kg, and 10,000 mg/kg, respectively. No abnormality and death were observed in mice of each group in 14 days. Also, no difference was detected in the mice body weight (Supplementary Fig. 9a) and main organs, including heart, liver, spleen, lung, and kidney, between drug-treated and control groups (Supplementary Fig. 9b, c). These data confirmed the favorable safety properties of YK-2-69 in vivo.

To further explore the pharmacokinetic profiles of YK-2-69, the Sprague-Dawley (SD) rats ($n = 3$/group) were administrated YK-2-69 by oral and intravenous injection (Table 1). In the intravenous group, the half-life ($t_{1/2}$), $C_{max}$, and $AUC_{0-\infty}$ values were 3 h, 974 ng/mL, and 1503 h·ng/mL, respectively. In oral administration group, YK-2-69 displayed the pharmacokinetic parameters as follows: $t_{1/2} = 5$ h, $C_{max} = 674$ ng/mL, and $AUC_{0-\infty} = 8384$ h·ng/mL. Moreover, the oral bioavailability of YK-2-69 is 56%. In summary, these results demonstrated the favorable druggability of YK-2-69 with favorable safety properties and pharmacokinetic profiles in vivo.

**YK-2-69 displayed more potent suppression on PCa than first-line drugs enzalutamide and palbociclib in vivo.** To evaluate antitumor activities of YK-2-69 in vivo, the DU145 xenograft mouse model was first established. Enzalutamide, the first-line PCa drug, and palbociclib, the selective CDK4/6 inhibitor in the market, were selected as the positive controls since CDK4/6 are down-regulated in DU145 cells when treated by YK-2-69. They were administered orally once a day for seven consecutive weeks. The low dose of YK-2-69 (100 mg/kg) displayed similar anti-tumor activities with enzalutamide but better activities than palbociclib. While the high dose of YK-2-69 (200 mg/kg) demonstrated much better antitumor activities than both enza-lutamide and palbociclib (Fig. 7a, c, d). Especially, different from enzalutamide and palbociclib which only delayed the tumor growth, the high dose of YK-2-69 not only suppressed the growth of tumor, but also decreased the volume of tumor since the 31st day (Fig. 7a). It is noteworthy that the body weight of mice also increased gradually in the high dose group (Fig. 7b). H&E

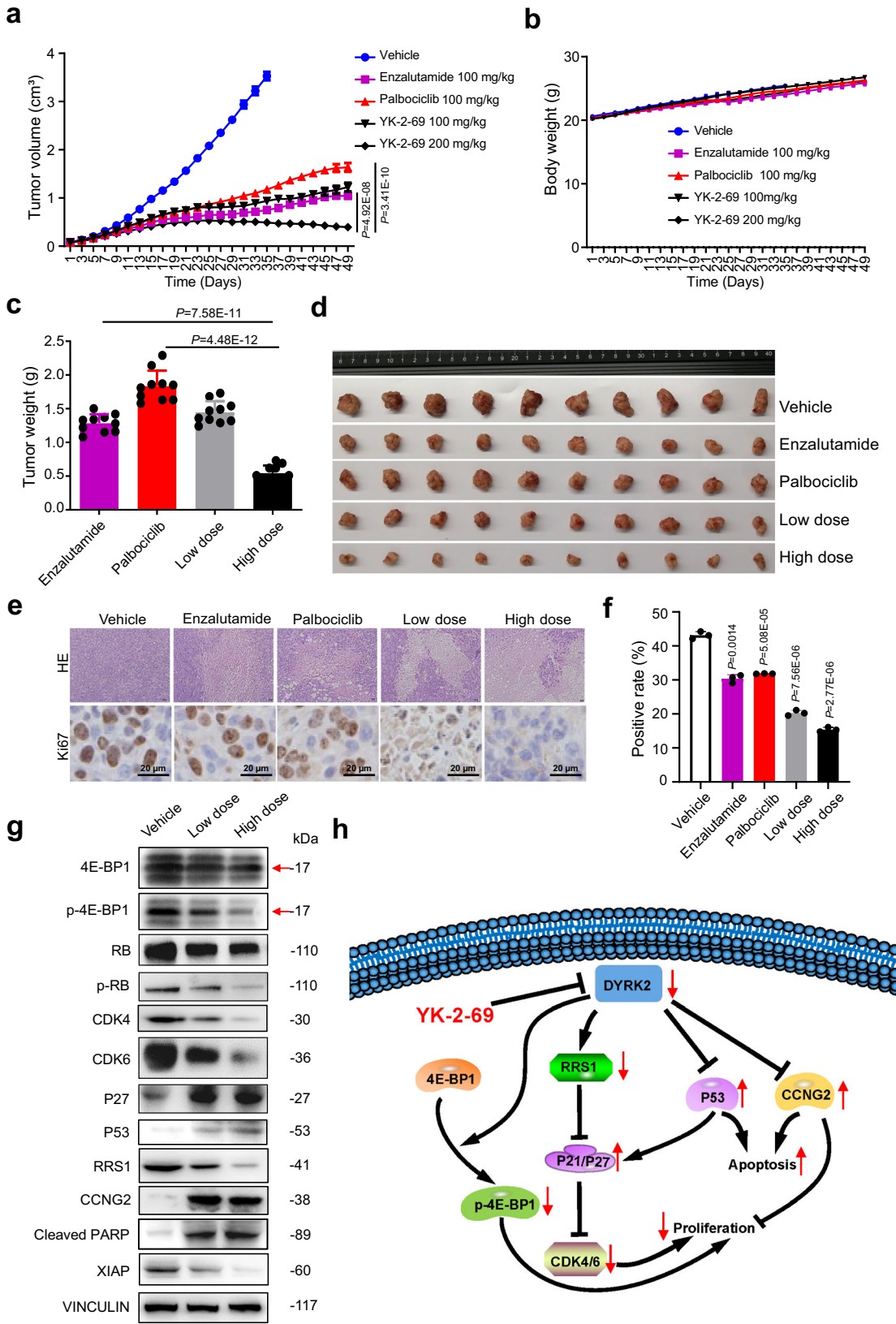

staining of tumor tissues and immunohistochemical analysis of Ki-67 expression indicated that YK-2-69 exhibited potent efficacy in killing PCa cells and inhibiting cell proliferation (Fig. 7e, f). Similar to the in vitro data, the WB analysis of the tumor tissue excised from DU145 xenograft mouse also indicated that the cell proliferation and apoptosis-related proteins p-4E-BP1, p-RB, CDK4/6, RRS1, and XIAP were deregulated, and P27, P53, CCNG2, and cleaved PARP were upregulated (Fig. 7g). Meanwhile, YK-2-69 also significantly inhibited tumor growth in the PC-3 xenograft mouse model. Furthermore, same as YK-2-69 reducing tumor size in the DU145 xenograft model, the high dose of YK-2-69 could also decrease tumor volume in the PC-3

**Fig. 7 YK-2-69 demonstrated remarkable antitumor activities in vivo. a–d** BABLc nude mice received subcutaneous injection of $1 \times 10^7$ DU145 cells in the right flank. When tumors grew ~80–100 mm$^3$, mice ($n = 10$/group) were orally administrated vehicle, palbociclib (100 mg/kg), enzalutamide (100 mg/kg), and YK-2-69 (100 and 200 mg/kg) every day. Tumor volumes (**a**) and body weight of mice (**b**) were measured every 2 days. After 35 days, mice in the control group were killed. After 49 days, mice of treatment groups were killed. Tumor tissues of each group were weighed (**c**) and then photographed (**d**). Unpaired two-tailed Student's $t$ test. Error bar, mean ± SD, $n = 10$. **e** Representative images of H&E and Ki-67 staining of paraffin section of tumor from mice treated with vehicle, palbociclib, enzalutamide, and YK-2-69. **f** Quantification of Ki67 positive rate of tumor from mice treated with vehicle, palbociclib, enzalutamide, and YK-2-69. Unpaired two-tailed Student's $t$ test. Error bar, mean ± SD, $n = 3$. **g** The total proteins in the tumor were extracted and used in the western blotting analysis of indicated proteins. The special bands of 4E-BP1 and p-4E-BP1 were shown with arrows. **h** A proposed model for inhibition of DYRK2 by YK-2-69 for the treatment of PCa. Source data are provided as a Source Data file.

xenograft model (Supplementary Fig. 10). Taken together, YK-2-69 displayed much better antitumor activity than first-line drugs enzalutamide and palbociclib in vivo.

Based on these results, YK-2-69 selectively binds to DYRK2 and inhibits its kinase activity to suppress the phosphorylation of 4E-BP1, which results in the inhibition of cell proliferation (Fig. 7h). The inhibition of DYRK2 by YK-2-69 can also down-regulate RRS1 and further up-regulate P21 and P27 to suppress CDK4/6[51,52]. RRS1-P21/27-CDK4/6 axes can restrain the transition from G1 to S phase of the cell cycle and eventually inhibit cell proliferation. Simultaneously, down-regulation of DYRK2 by YK-2-69 can increase the P53 and CCNG2 level, which promotes apoptosis[55,56]. In summary, this highlights DYRK2 inhibition by YK-2-69 as a promising combination to inhibit proliferation and promote apoptosis, which provides a "kill two birds with one stone" regimen of PCa.

## Discussion

The expression level of DYRK2 widely depends on the human tumor tissues, and it plays diverse roles in the occurrence and development of various cancers, which highlights the possibility of DYRK2 as a potential target for cancer treatment. Previous reports mainly considered DYRK2 as a cancer suppressor, which induces apoptosis through promoting phosphorylation of P53 in colorectal cancer[65], inhibits cell cycle from G1 to S phase via degrading c-JUN and c-MYC in breast cancer[22], and suppresses EMT by accelerating SNAIL degradation in glioma[66]. But not all reports demonstrate that DYRK2 inhibits cancer initiation and growth as a cancer suppressor. Recent reports revealed the DYRK2 accelerated G1 to S phase transition of the cell cycle through regulating 26S proteasome activity, and was considered as an oncogene in MM and TNBC[26,31]. Considering the critical function of DYRK2 in cancers, we conducted a data-mining of TCGA to find the high expression of DYRK2 in PCa, which was positively correlated with clinical prognosis and mortality (Fig. 1a–d). However, the function of DYRK2 in PCa is still unclear. In our work, DYRK2 was found to be highly expressed in PCa patient samples and cells (Fig. 1e–g). The knock-down of DYRK2 in PCa cells significantly inhibited cell growth and metastasis, caused a G1 arrest of the cell cycle, and induced apoptosis (Fig. 2c–h and Supplementary Fig. 2b–d). In addition, knock-down of DYRK2 significantly inhibited tumor growth in vivo (Fig. 2j and Supplementary Fig. 2h). All these results indicate that DYRK2 is a potential therapeutic target for the treatment of PCa.

Although several DYRK2 inhibitors were discovered and exhibited anti-cancer activity in MM and TNBC[31,32], their selectivity over DYRK family members and drug-like properties need to be further modified. To develop more potent and selective DYRK2 inhibitors, we performed a high-throughput virtual screening and identified a DYRK2 hit with a benzothiazole chemical scaffold, which was further modified to offer the highly selective inhibitor YK-2-69. YK-2-69 exhibited stronger inhibition to DYRK2 with an IC$_{50}$ value of 9 nM and showed selectivity

over the DYRK subfamily and a panel of 370 kinases (Fig. 3). To explore the clear mechanism and elucidate the exact interaction of YK-2-69 with DYRK2, we solved their co-crystal structure with a high resolution at 2.5 Å (PDB ID: 7EJV), which showed the essential interaction residues Lys-231 and Asn-234 (Fig. 4).

Similar to the knock-down of DYRK2 in the PCa cells, YK-2-69 also significantly inhibited cell growth through G1 arrest and apoptosis induction, and decreased the EMT activity (Fig. 5 and Supplementary Fig. 7). Transcriptome-wide RNA sequencing assays demonstrated that DYRK2 KD and YK-2-69 treatment played important and similar roles in cell proliferation inhibition (Fig. 6 and Supplementary Fig. 8). Importantly, YK-2-69 displayed acceptable safety properties with a maximal tolerable dose of >10,000 mg/kg (Supplementary Fig. 9) and favorable pharmacokinetic profiles with 56% bioavailability (Table 1) in vivo. Moreover, YK-2-69 exhibited much better antitumor activities than both enzalutamide and palbociclib. Especially, YK-2-69 not only suppressed the growth of tumor, but also decreased the volume of tumor, which was completely different from enzalutamide and palbociclib (Fig. 7 and Supplementary Fig. 10). These results provided us a possibility that YK-2-69 may contribute to solving the drug-resistant dilemma of enzalutamide as hormonal therapy. YK-2-69 exhibited much higher anti-PCa efficacy via synergistic regulation on a panel of pathways, including DYRK2-4E-BP1, DYRK2-RRS1-P21/27-CDK4/6, and so on, to promote apoptosis and inhibit proliferation. This might be one of the possible reasons why YK-2-69 exhibited significant anti-PCa efficacy.

The latest data from International Agency for Research on Cancer reports (World Cancer Report 2020) that prostate cancer is the second most common cancer in men worldwide, with an estimated 1.3 million new cases and 360,000 deaths in 2020. However hormonal therapy, the leading treatment of PCa, is only a remission but not a cure for PCa, and most PCa patients became resistant to hormonal therapy at last. Therefore, it is urgent and meaningful to identify novel targets and develop new drugs for PCa. Our work identified DYRK2 as a potential drug target and verified its critical roles in PCa, which offers a valuable direction for the treatment of PCa. Especially, we discovered a highly selective DYRK2 inhibitor with favorable druggability, which could be used as a small-molecule probe for biological studies and also provide a potential candidate for PCa clinical treatment. Since DYRK2 plays critical roles in various human cancers, targeting DYRK2 could also provide a promising opportunity for other patients with refractory cancers.

## Methods

**Protein expression and purification.** DYRK2 $^{72–479}$ with an N-terminal 6× His affinity tag and TEV protease cleavage site was cloned into the pET28a vector. Sequence verified plasmid was transformed into *E. coli* BL21 (DE3) cell (Weidibio, Cat#: EC1002). Bacterial cultures were grown at 37 °C in LB medium to an OD$_{600}$ of 0.8 before being induced with 0.5 mM isopropyl β-D-1-thiogalactopyranoside (IPTG, Inalco, Cat#: 1758-1400) overnight at 25 °C. Cells were collected by centrifugation and suspended in the lysis buffer containing 20 mM HEPES, pH 7.5, 500 mM NaCl, 20 mM imidazole, and 10% glycerol, and disrupted by sonication.

The lysate was centrifuged at 40,000 g for 30 min twice at 4 °C. After centrifugation, the supernatant was initially purified using a Ni-NTA column (GE Healthcare, Cat#: 17-5318-03), and then eluted with lysis buffers supplemented with 300 mM imidazole. Eluent recombinant protein with the His$_6$-tag was subsequently cleaved using TEV protease at 4 °C overnight. The cleaved protein was further purified using reverse affinity chromatography and size-exclusion chromatography in buffer containing 20 mM HEPES, pH 7.5, 250 mM NaCl. The pure protein was concentrated to 11.2 mg/mL and flash-frozen with liquid nitrogen for later usage.

**Co-crystallization, data collection, and structure determination.** DYRK2 $^{72–479}$ (10 mg/mL) was incubated with 1 mM YK-2-69 at 4 °C before crystallization. The protein-YK-2-69 mixture was then mixed in a 1:1 ratio with crystallization solution (0.1 M sodium citrate pH 5.5, 8% PEG3350) in a final drop size of 2 μL. The initial DYRK2-YK-2-69 crystals were grown at 4 °C by the hanging-drop vapor diffusion method and optimized by seeding. Cuboid-shaped crystals appeared after 2–7 days. Crystals were cryoprotected in the crystallization solution supplemented with 30% glycerol before being frozen in liquid nitrogen. The X-ray diffraction data were collected using a PILTUS3 6 M detector on beamline BL19U at Shanghai Synchrotron Radiation Facility (SSRF, Shanghai, China)[67]. The diffraction data were indexed, integrated, and scaled using HKL-2000. The structure was determined by molecular replacement using the published DYRK2 structure (PDB ID: 5LXD) as the search model using the Phaser-MR program in Phenix[47,68]. A clear electron density was observed in the center of the ATP binding pocket after molecular replacement. YK-2-69 was fitted using the eBLOW and LigandFit program in Phenix[69,70]. The structural model was further adjusted in Coot and refined using Phenix while using NCS restrains[71,72]. The quality of the structural model was checked using the MolProbity program in Phenix[73]. The crystallographic data and refinement statistics are summarized in Supplementary Table 2.

**Clinical samples.** Three pairs of prostate tumor and matched normal tissues were obtained from Huai'an First People's Hospital with patients' informed consent. The human samples used in this study were approved by Ethics Committee of Huai'an First People's Hospital. Clinicopathologic information of these three patients was listed in Supplementary Table 3.

**Immunohistochemistry of prostate cancer patient samples.** PCa tissues and adjacent normal prostate tissues were fixed in 4% formaldehyde solution and processed routinely for paraffin embedding. Sections were cut at around 4 μm thickness and placed on glass slides, and stained with DYRK2 antibody. Add 2 drops of freshly prepared DAB solution to each sheet, and then re-dyeing and dehydration seal. Sections were scanned by a digital pathology scanner.

**Cell culture.** The PCa DU145, 22Rv1, and LNCaP cell lines were obtained from American Type Culture Collection (ATCC), and cultured in endotoxin-free RPMI1640 supplemented with 10% fetal bovine serum (FBS, Gibco, Cat#: 10099141C)[74]. HEK-293T human embryonic kidney cells (ATCC) were cultured in DMEM with 10% FBS. The DU145 shNC/shDYRK2 and 22Rv1 shNC/shDYRK2 cells were also cultured in endotoxin-free RPMI1640 with 10% FBS. PC-3 human PCa cells (ATCC), were cultured in F-12K with 10%FBS. RWPE-1 normal epithelial prostatic cells (ATCC) were cultured in KM supplemented with 10% FBS[75]. All the cells are not among commonly misidentified cell lines, and were tested for mycoplasma contamination annually using a PCR Mycoplasma Detection Kit (Applied Biological Materials Inc., Cat#: G238). To prevent potential contamination, all the media were supplemented with Penicillin-Streptomycin (Beyotime, Cat#: C0222) and Plasmocin prophylactic (InvivoGen, Cat#: ant-mpp) according to the manufacturer's instructions.

**Anti-proliferation activity assays.** We seeded 5,000 cells/well, PCa cells on a 96-well plate and treated with DMSO or tested compounds for 72 h. Add 10 μL of CCK8 reagent (Share-bio, Cat#: SB-CCK8) to each well, mix lightly, and incubate the plates in an incubator at 37 °C with 5% CO$_2$ for 1–4 h. The incubated cell culture plate was placed on the enzyme plate analyzer (Bio-Tek, Cat#: SynergyH1) and the absorbance value was measured at 450 nm. In all, 2000 cells/well PCa cells were seeded and subjected to DYRK2 inhibitor YK-2-69 treatment for 5 days and the cell culture plate was placed on the enzyme plate analyzer every day and the absorbance value was measured at 450 nm.

**Cell growth assays.** In all, 2000 cells/well PCa cells were seeded and subjected to DMSO or tested compounds treatment for 5 days and the cell culture plate was placed on the enzyme plate analyzer every day and the absorbance value was measured at 450 nm.

**Lentivirus production and infection.** Lentivirus-induced DYRK2 KD was modified in DU145 or 22Rv1 cells. In brief, 0.5 μg pMD2.G, 0.3 μg pMDLg/pRRE, and 0.7 μg PrSV-Rev, and 1.5 μg pLKO-shDYRK2 or pLKO-shNC (Genechem Co. Ltd. Shanghai, China) were co-transfected into 293T cells in cell culture dish. The effectene transfection reagent packs lentiviruses. Lentivirus particles were collected at 48 h and 72 h after transfection and transferred directly into DU145 and 22RV1 cells containing 4 μg/mL polypropylene. PCa cells, including lentiviruses, were then rotated and inoculated for 90 min at 32 °C and 135 g. Finally, 1 μg /mL puromycin was added to cultured PCa cells 48 h after rotation inoculation to select positive infected cells. The shRNA targeting oligo sequence: CCGGGCAGGGT AGAAGCGGTATTAACTCGAGTTAATACCGCTTCTACCCTGCTTTTTG.

**Real-time quantitative PCR.** DYRK2 KD cells or PCa cells were treated with vehicle or YK-2-69 at indicated concentrations for 72 h. Above the cells and prostate cancer patient tissue cells with total RNA isolated with the TRIZOL reagent (Thermo Fisher Scientific, Cat#: 15596026) was subjected to reverse transcription using the PrimeScriptTM RT reagent Kit (RR047Q, Takara). Real-time quantitative PCR reactions were performed with the THUNDERBIRDSYBR qPCR Mix (QPS201, TOYOBO) and primers (Tsingke Biotechnology Co., Ltd., Beijing, China) listed in Supplementary Table 4.

**Immunoprecipitation.** We lysed the cells in PBS containing 20% Triton X-100, 10% CHAPS, and Protease Inhibitor Mixture (Roche, Cat#: 04693132001) for 30 min at room temperature. After centrifugation for 5 min, the supernatant was incubated with 10 μL of anti-DYRK2 antibody (Abcepta, Cat#: AP7534a, 1:1000) for 2 h at 4 °C. Rabbit IgG (Beyotime, Cat#: A7016, 1:200) was used as a control. Incubate with antibody and supernatant with 2% BSA and 10 mg protein A-Sepharose beads (Sigma-Aldrich, Cat#: P1406) at 4 °C overnight. On the next day, the protein was eluted three times with 0.1% PBST, resuspended in 2× SDS-PAGE loading buffer, and boiled for 5 min. The eluate was fractionated by SDS-PAGE.

**Western blotting analysis.** DU145, PC-3, 22Rv1, LNCaP, and RWPE-1 cells were grown in T-75 flasks at $5 \times 10^6$ cells/mL. PCa cells were treated with vehicle or the specified YK-2-69 concentration for 72 h, and treated cells were harvested and lysed by sonication in receptor lysis buffer (RLB) containing 20 mM HEPES (pH 7.5), 500 mM NaCl, 1% Triton X-100, 1 mM DTT, 10% glycerol, phosphatase inhibitors (50 mM NaF, 1 mM Na$_3$VO$_4$), and protease inhibitor mix. Lysates from cells and tumor tissues were quantitated and 20 to 50 μg of protein lysates were boiled in an SDS sample buffer, size fractionated by SDS-PAGE, and transferred onto a PVDF membrane (Immobilon). After blocking in 5% nonfat dry milk (or 3% BSA), membranes were incubated with the following primary antibodies overnight: DYRK2 rabbit polyclonal antibody (Abcepta, Cat#: AP7534a, 1:1000), RB rabbit polyclonal antibody (Proteintech, Cat#: 10048-2-Ig, 1:5000), Phospho-Rb (Ser807/811) rabbit monoclonal antibody (Cell Signaling Technology, Cat#: 8516, 1:1000), CDK4 rabbit monoclonal antibody (Cell Signaling Technology, Cat#: 12790, 1:1000), CDK6 mouse monoclonal antibody (Cell Signaling Technology, Cat#: 3136, 1:2000), PARP rabbit monoclonal antibody (Cell Signaling Technology, Cat#: 9532, 1:1000), Cleaved PARP rabbit monoclonal antibody (Beyotime, Cat#: AF1567, 1:1000), RRS1 rabbit polyclonal antibody (Proteintech, Cat#: 15329-1-AP, 1:1000), CCNG2 rabbit polyclonal antibody (Abcam, Cat#: ab251826, 1:500), P21 rabbit polyclonal antibody (Proteintech, Cat#: 10355-1-AP, 1:1000), P27 rabbit monoclonal antibody (Cell Signaling Technology, Cat#: 3686, 1:1000), P53 rabbit monoclonal antibody (Cell Signaling Technology, Cat#: 2527, 1:1000), XIAP rabbit polyclonal antibody (Proteintech, Cat#: 10037-1-AP, 1:1000), E-cadherin rabbit polyclonal antibody (Proteintech, Cat#: 20874-1-AP, 1:5000), 4E-BP1 rabbit monoclonal antibody (Cell Signaling Technology, Cat#: 9644, 1:1000), Phospho-4E-BP1 (Thr37/46) rabbit monoclonal antibody (Cell Signaling Technology, Cat#: 2855, 1:1000), Vinculin rabbit polyclonal antibody (Proteintech, Cat#: 26520-1-AP, 1:1000), Alpha Tubulin mouse monoclonal antibody (Proteintech, Cat#: 66031-1-Ig, 1:20000). Following three washes in PBST, the blots were incubated with secondary antibody Goat Anti-Mouse IgG, H&L Chain Specific Peroxidase Conjugate (Merck, Cat#: 401215-2 ML, 1:5000), or Goat Anti-Rabbit IgG, H & L Chain Specific Peroxidase Conjugate (Merck, Cat#: 401315-2 ML, 1:5000). Proteins were detected by chemiluminscent detection system (Tanon, Shanghai, China) and analyzed by Image J software.

**Colony formation assays.** Colony formation assays were performed with $5 \times 10^2$ cells, which were plated to a 24-well plate. Two weeks after initial plating, cells were fixed, stained with 0.1% crystal violet (Beyotime, Cat#: C0121) and counted.

**Migration and invasion assays.** Cells were treated with vehicle or different concentrations of tested compounds for 48 h, and equal numbers ($5 \times 10^4$ cells per well) of the cells were seeded in FBS-free RPMI-1640 culture medium in the presence of vehicle or different concentrations of tested compounds in the upper chambers of 8-μm pore size transwell inserts. The lower chambers were filled with 500 μL of medium supplemented with 10% FBS. Cells were allowed to invade the bottom chamber for 24 h. Non-invading or non-migrating cells in the upper surface were removed, and invaded or migrated cells on the lower surface were fixed with 4% paraformaldehyde and stained with 0.1% crystal violet for 5 min. The stained cells were photographed and quantified.

**Cell cycle and apoptosis assays.** In this study, propidium iodide (PI) DNA staining kit (Beyotime, Cat#: C1052) was chosen to assess the cells located at G0/

G1, S, and G2/M stages. For PI staining, $1 \times 10^6$ cells were collected, washed once with PBS, 70% ethanol was added, gently beaten, fixed at 4 °C for 12 h, centrifuged at $1000 \times g$ for 5 min, and the cells were precipitated. Carefully suck out the supernatant and add 1 mL PBS and suspended in 0.5 mL Krishan's buffer supplemented with 0.05 mg/mL PI, 0.1% trisodium citrate, 0.02 mg/mL ribonuclease A, and 0.3% NP-40, incubated at 37 °C for 30 min and then applied to flow cytometer (FACSVerse, BD, USA) directly. The samples were transferred onto the ice before being subjected to flow cytometry. Cell apoptosis was assayed by Annexin V-APC and PI apoptosis kit (Elabscience, Cat#: E-CK-A217). Cells were seeded at $1 \times 10^6$/well in 10% FBS–RPMI1640 into six-well plates and treated with tested compounds for 24 h. The cells were then washed twice with cold PBS and resuspended in 1×binding buffer (0.1 M HEPES (pH 7.4), 1.4 M NaCl, 25 mM CaCl$_2$) at a concentration of $1 \times 10^6$ cells/mL. A 100 µL volume of the solution ($1 \times 10^5$ cells) was transferred to a 5-mL culture tube; 5 µL of Annexin V-APC and 5 µL PI were added to each tube. The cell suspension was gently vortexed and incubated for 30 min at room temperature (25 °C) in the dark, and then 200 µL PBS was added to each tube. The apoptosis assay was carried out by flow cytometry at 633 nm excitation, and the results were analyzed with FlowJo V10 software. Supplementary Fig. 11 and 12 exemplified the gating strategy in analyzing the results of cell cycle and apoptosis assays, respectively.

**DYRKs kinase activity and kinase-inhibitor specificity profiling assays**. For evaluating the inhibitory effect of tested compounds on DYRKs activity, we first prepared reaction buffer containing 20 mM HEPES (pH 7.5), 10 mM MgCl$_2$, 1 mM EGTA, 0.01% Brij35, 0.02 mg/mL BSA, 0.1 mM Na$_3$VO$_4$, 2 mM DTT, and 1% DMSO. Subsequently, DYRKs kinase and DYRKtide substrate were added into the reaction buffer and mixed gently. After tested compounds were delivered into the reaction mixture, $^{33}$P-ATP (specific activity 0.01 µCi/µL final) was added to initiate the reaction. Then, the reactions were spotted onto P81 ion exchange paper (Whatman, Cat#: 3698-915) followed by incubation for 2 h at room temperature. The filters were washed extensively in 0.75% phosphoric acid. Finally, measure the radioactive phosphorylated substrate remaining on the filter paper and calculate the remaining DYRKs activities in the tested compound group relative to the DMSO group.

Kinase inhibitor specificity profiling assays were carried out at Reaction Biology Corporation (https://www.reactionbiology.com/). YK-2-69 kinase specificity was determined against a panel of 370 protein kinases at the concentration of 1 µM. YK-2-69 was added to the mixture of the indicated kinase and substrate solution, then $^{33}$P-ATP was added to initiate the reaction. After 120 min, the reaction mixture was spotted onto P81 ion exchange paper, then washed in 0.75% phosphoric acid. At last, measuring the radioactive phosphorylated substrate remaining on the filter paper to determine the kinase activity.

**Microscale thermophoresis binding assay**. Binding affinities of compounds with the DYRK2 protein were measured by using the Monolith NT.115.The DYRK2 protein was kept in the PBS-P buffer at a concentration of 10 µM and then labeled according to the protocol of Protein labeling kit RED-NHS 2nd Generation (Nanotemper, Cat#: MO-L011). The compounds at a range of concentrations were incubated with labeled DYRK2 at room temperature for 10 min in the dark. The mixtures were loaded into 16 hydrophilic glass capillaries, then binding affinities were measured by monitoring the thermophoresis with 40% LED power on Monolith NT.115.The data were analyzed by Mo.Affinity Analysis v2.2.4 software.

**RNA-seq sequencing**. Total RNA samples were isolated from the YK-2-69 or enzalutamide treated and DYRK2-KD cells respectively. RNA concentration was measured by NanoDrop 1000, and RNA integrity was measured by 2100 BioAnalyzer. According to manufacturer's instructions, adding an appropriate amount of MIX1 or MIX2 to each RNA sample. Using the Kapa chain mRNA-seq kit (Illumina) (Kapa Biosystems, Cat#: KK8541), a library of 300 ng total RNA was constructed for each sample through 10 PCR amplification cycles. The library was purified using the AxyPrep MAG PCR Normalizer kit (Axygen, Cat#: MAG-PCR-NM-50). Each library was quantified using a Qubit fluorometer and assessed for size distribution using 2100 BioAnalyzer. Sequencing was performed on Illumina HiSeq 2500 apparatus with a V4 chemically generated 51 bp single-ended read sequence. Each group contains 3–4 repeats and the corresponding control group keep the same number of repeats. Quality control of RNA-Seq reads was performed using FASTQC. The reads with low complexity or low quality were removed using Cutadapt. Trimmed reads were aligned to human genome reference (GRCh38) using STAR, and uniquely mapped reads were retained in the downstream analysis. RSEM was used to calculate the expression levels of genes. DEG analysis was performed using DESeq2, $P < 0.05$, and 2-fold change was used as statistical significance. Hierarchical clustering analysis was performed using the R package 'mclust'. GSEA was employed to calculate enrichment pathways based on the signature gene sets from the Molecular Biology Database (MSIGDB).

**Structural optimization**. To obtain more potent DYRK2 inhibitor, structural optimization on the lead compound **12** was conducted (Supplementary Fig. 4). Firstly, the isopropyl group on the piperazine was replaced with different sizes substituents (**16-18**), among which compounds **16** and **18** with a small ethyl or

hydrogen group showed similar activity as lead **12**, while compound **17** with a big Boc group exhibited significantly decreased activity. When benzothiazole was changed to pyrazolo[1,5-a]pyrimidine (**19**), DYRK2 inhibitory activity significantly decreased. Therefore, benzothiazole was retained in the following structural optimization. To our delight, the optimization on the linker between pyridine and piperazine (**20**: 35 nM and **21**: 85 nM) remarkably improved DYRK2 inhibitory activity, especially the carbonyl linker. Subsequently, the change of pyridine to pyrimidine (**22**: 697 nM) also decreased DYRK2 inhibitory activity. Therefore, pyridine is a favorable moiety for the maintenance of activity. After confirming the benzothiazole, pyridine, and carbonyl linker as the optimized groups, the subsequent modification was carried out by introducing substituents on the benzothiazole. The introduction of methyl substituent generated **23** and **24**, which exhibited stronger inhibition on DYRK2 with IC$_{50}$ values of 22 and 27 nM, respectively. The introduction of dimethylamine substituent (**25**, **26**) further improved DYRK2 inhibitory activity, and **26** exhibited the most potent inhibition on DYRK2 with an IC$_{50}$ value of 9 nM. The change of pyrimidine to thieno[3,2-d] pyrimidine generated **27**, which showed no inhibition on DYRK2. Taken together, **26** was confirmed as the most potent DYRK2 inhibitor, which was re-named as YK-2-69 and selected for further biological evaluation.

**Acute toxicity studies**. The experimental procedures and animal use and care protocols were approved by the Institutional Animal Care and Use Committee (IACUC) of China Pharmaceutical University. To study the safety in vivo, seven-week ICR mice (weight 18–22 g), purchased from Shanghai SLAC Laboratory Animals Co. Ltd., half male and half female, were randomly divided into one control group and three treatment groups ($n = 10$/group). The temperature and humidity of the animal room is 20–26 °C and 40–70%, respectively. All mice were given 12 h of light and 12 h of darkness in turn each day. Mice of treatment groups were administrated by oral at a dose of 2500, 5000, and 10,000 mg/kg, respectively. After single dose, the signs of toxicity were observed, and body weight was recorded once 2 days in 14 days.

**Pharmacokinetic profiles**. The experimental procedures and animal use and care protocols were approved by the Institutional Animal Care and Use Committee (IACUC) of China Pharmaceutical University. SD rats, purchased from Shanghai SLAC Laboratory Animals Co. Ltd., were used to determine the pharmacokinetic profiles of YK-2-69. SD rats were divided into intravenous and oral administration groups ($n = 3$/group). The temperature and humidity of the animal room is 20–26 °C and 40–70%, respectively. All mice were given 12 h of light and 12 h of darkness in turn each day. The dose of intravenous and oral administration groups was 1 and 10 mg/kg, respectively. Blood samples of the intravenous group were collected at 2 min, 5 min, 15 min, 30 min, 1 h, 2 h, 4 h, 6 h, 8 h, and 12 h. and blood samples of the oral administration group were collected at 5 min, 15 min, 30 min, 1 h, 2 h, 4 h, 6 h, 8 h, 12 h, and 24 h. The concentrations of YK-2-69 in serum were measured by LC/MS/MS.

**In vivo antitumor activity**. The experimental procedures and animal use and care protocols were approved by the Institutional Animal Care and Use Committee (IACUC) of China Pharmaceutical University. The temperature and humidity of the animal room is 20–26 °C and 40–70%, respectively. All mice were given 12 h of light and 12 h of darkness in turn each day. BALB/c nude mice were purchased from Shanghai SLAC Laboratory Animals Co. Ltd.

BALB/c nude mice received subcutaneous injection of $1 \times 10^7$ DU145 shNC and shDYRK2 cells in the right flank to establish shNC group ($n = 6$) and shDYRK2 group ($n = 10$), respectively. After the formation of tumors, tumor volumes and body weight were measured once every two days. After 29 days, the shNC group mice were killed for humane reasons, and tumor tissues were weighed and taken photos. After 49 days, shDYRK2 group mice were killed, and tissues were weighed and taken photos. Tumor tissues of each group were kept at −80 °C for further analysis.

BALB/c nude mice received subcutaneous injection of $1 \times 10^7$ 22Rv1 shNC and shDYRK2 cells in the right flank to establish shNC group ($n = 6$) and shDYRK2 group ($n = 8$), respectively. After the formation of shNC group mice tumors, tumor volumes of shNC group were measured once every two days, and no visible tumors were detected in the shDYRK2 group mice. Meanwhile, body weight of shNC group and shDYRK2 group were measured once every two days. After 19 days, the mice were killed, and tumor tissues were weighed and taken photos.

BALB/c nude mice received subcutaneous injection of $1 \times 10^7$ DU145 cells in the right flank. When the average tumors reached the volumes of 80–100 mm$^3$, the mice were randomly divided into control ($n = 10$/group), drug treatment groups ($n = 10$/group). YK-2-69 at doses of 100 and 200 mg/kg were given by oral every day. Palbociclib and enzalutamide were given to mice orally at a dosage of 100 mg/kg/d and used as positive references for comparison, while control mice received an equal volume of saline. Tumor volumes and body weight were measured once every 2 days. After 35 days, the mice in the control group were killed for humane reasons, and tumor tissues were weighed and taken photos. After 49 days, mice of treatment groups were killed, and tumor tissues were weighed and taken photos. Tumor tissues of each group were kept at −80 °C for further analysis.

BALB/c nude mice received subcutaneous injection of $1 \times 10^7$ PC-3 cells in the right flank. When the average tumors reached the volumes of 80–100 mm³, the mice were randomly divided into control, palbociclib (100 mg/kg), and YK-2-69 (100 or 200 mg/kg) groups ($n = 8$/group), and they were given by oral every day. Tumor volumes and body weight were measured once every two days. After 29 days, mice were killed, and tumor tissues were weighed and taken photos. Tumor tissues of each group were kept at $-80\,°C$ for further analysis.

**H&E and Ki67 Staining**. Tumor tissues and normal tissues were fixed in 4% formaldehyde solution and processed routinely for paraffin embedding. Sections were cut at around 4 μm thickness and placed on glass slides, and counterstained with hematoxylin and eosin and anti-Ki-67.

**Statistics and reproducibility**. Data were expressed as mean ± SD. Statistical analysis was performed using GraphPad Prism 8 software. For experiments with two groups, two-tailed Student's $t$ test was used. $P < 0.05$ was considered to be statistically significant. As indicated in the figure legends, all in vitro experiments were performed in three biological replicates unless stated otherwise. Representative micrographs and western blot shown in figures were repeated three times independently with similar results.

**Reporting summary**. Further information on research design is available in the Nature Research Reporting Summary linked to this article.

## Data availability

The DYRK2-YK-2-69 complex in this study has been deposited in the Protein Data Bank under accession code 7EJV. The cited DYRK2-EHT 1610 and DYRK2-LDN192960 complex in this study can be found in the Protein Data Bank under accession code 5LXD and 6K0J, respectively. The raw RNA-seq data generated in this study have been deposited in the BIG Data Center under the accession number: HRA002200 (RNA-seq data in DU145 cells) and HRA002197 (RNA-seq data in 22Rv1 cells). Source data are provided with this paper.

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

## Acknowledgements

This study was supported by National Natural Science Foundation of China (82073701, 31970547, and 31900687), Natural Science Foundation of Jiangsu Province (BK20190552, BK2019040713) and the Project Program of State Key Laboratory of Natural Medicines, China Pharmaceutical University (SKLNMZZ202013, SKLNMZZRC201808, SKLNMZZ202014, and SKLNMKF202004). The X-ray data were collected at the Shanghai Synchrotron Radiation Facility (SSRF) BL19U beamline.

## Author contributions

K.Y., Z.L., W.K., and X.W. conceived the project, which had leadership from Y.J., H.H., Y.X., and P.Y. All authors contributed to manuscript writing, review, and editing. K.Y., W.K., X.W., and J.D. contributed to cancer database analysis. K.Y., Z.L., W.K., X.W., M.J., W.C., J.D., J.L., L.W., and Y.J. contributed to biological experiments, including immunohistochemistry, cell proliferation assays, lentivirus production and infection, real-time quantitative PCR, immunoprecipitation, western blotting, colony formation assays, migration and invasion assays, cell cycle and apoptosis assays, RNA-seq sequencing, etc. K.Y., W.M., and H.G. contributed to structure-based virtual screening. K.Y., W.C., C.S., and P.Y. contributed to the design and synthesis of DYRK2 inhibitors. Z.L., X.Y., M.L., and Y.X. contributed to crystallography experiments. K.Y., H.H., and P.Y. contributed to in vivo acute toxicity properties, pharmacokinetic profiles, and antitumor activity evaluation. All authors contributed to data analysis and interpretation.

## Competing interests

The authors declare no competing interests.
