## [Peer review file · Nature Communications]

REVIEWER COMMENTS

Reviewer #1 (Remarks to the Author):

In the current paper entitled “targeting dual specificity tyrosine phosphorylation regulated kinase two (DYRK2) with a highly selective inhibitor for the treatment of prostate cancer,” Yuan et al., identified that DYRK2 is expressed in prostate cancer compared to normal tissue and that the genetic knock down of DYRK2 inhibits cancer cell proliferation migration and invasion in vitro and reduces tumor growth in vivo using a subcutaneous xenograft model. Further, the authors identify a novel inhibitor of DYRK2 that has good pharmacokinetics and is efficacious for the treatment of prostate cancer in vivo. While there is novelty to paper regarding the identification of DYRK2 and a YK-2-69, there are some serious concerns regarding the robustness of the findings given that the majority of the study is based on an androgen receptor negative prostate cancer cell line DU145. There were also additional major issues identified.

Major points.

- o The primary focus is based on DYRK2 expression in prostate cancer tissue vs. normal tissue. What about other family members, DYRK1B, DYRK3, DYRK4A and DYRK4B? Are these expressed in prostate cancer vs. normal tissue and do the other family members correlate with disease progression. There is also never an attempt to validate their findings in human tissues. Does the expression of DYRK correlate with MYC in humans?
- o For Figure 1. What is the correlation between DYRK2 and castrate resistant prostate cancer vs. castrate sensitive? The rationale for pursuing DYRK2 is based on 2-cell lines. PC3 represents small cell prostate cancer which is not reflective of adenocarcinoma and is AR negative. Further, DU145 is also androgen negative and derived from a brain metastasis. What is the effect of the drug on more representative prostate cancer cell lines such as 22Rv1/VCAP/C4-2 etc. This should be demonstrated for robustness of the studies.
- o Figure 2. Invasion and migration assay results may be confounded by the effects of DYRK2 knockdown on growth – less migration/invasion could be due to slower growth over 24 hours. The growth rate of the DYRK2 knockdown cells is remarkable given that DYRK2 is knocked down approx. by 75%. It looks as though they did not grow at all sub-cutaneously raising the questions as to what is being analyzed in Fig. 2H. What is the efficacy of DYRK2 knockdown on other prostate cancer cell lines in vivo.
- o Figure 4 – Need to show the effect of the YK-2-69 on multiple cell lines including RWPE. Specificity of YK-2-69 against other PCa cell lines in which DYRK2 is knocked down should be demonstrated. Further the effects of YK-2-69 on other cell lines should be demonstrated on additional PCa cell lines with regard to growth/migration.
- o Figure 5 is also solely based on DU145 with no validation of findings in other cell line models.
- o Fig. 6, remarkably, enzalutamide looks to have a significant effect on the AR negative DU145 cell line in vivo again raising concerns as to the robustness and rigor of the findings.

Minor Points

- o Enzalutamide and Palbociclib should be used as a control of in vitro data, especially fig 4.

o Some typos throughout. Line 73 “inhibits metastasis in vitro” Line 117 “inhibited cell cloning”

Statistics should be included for the in vivo studies.

Reviewer #2 (Remarks to the Author):

Manuscript ID: NCOMMS-21-19991-T

Title: Targeting dual-specificity tyrosine phosphorylation-regulated kinase 2 (DYRK2) with a highly selective inhibitor for the treatment of prostate cancer

The manuscript identifies DYRK2 as a novel drug target for prostate cancer and discover a highly selective DYRK2 inhibitor with favorable druggability for the treatment of prostate cancer (PCa). The manuscript is very complete as includes the demonstrations that DYRK2 is a potential therapeutic target for PCa, the identification and improvement of a highly selective inhibitor of DYRK2, the cocrystal structure of this inhibitor with DYRK2, the in vitro and in vivo activities of this compound and its pharmacokinetic parameters in rats. This manuscript could allow the development of new treatments for prostate cancer. The manuscript is well written, but it could be improved, specially thinking in a general reader of this multidisciplinary journal. For example:

- Figure 1 describes the higher expression of DYRK2 in PCa patients, the differences of expression of DYRK2 between patients with high or intermediate risk, differences in relapse-free survival, In this part, the mRNA level of DYRK2 were evaluated in 11 different cell lines, showing that PC-3 and DU-145 PCa cell lines showed much higher DYRK2 mRNA levels (Fig. 1f). But, which was the objective of analyzing these 11 different cell lines? All these are PCa cell lines? The addition of a sentence explaining a little more the main purpose of this experiment would be useful. In this figure, the *, ** and *** symbols represent the unpaired Student's t-test $p < 0.05$, $p < 0.01$ and $p < 0.001$, respectively; but which is the reference group? What does NA means?. In addition, figures 1a, 1b and 1c represent the values of DYRK2 expression measured in FPKM, but this unit is not defined. Also in figure 1g the protein levels of DYRK2 in PC-3 and DU-145 cell lines (why these two cell lines were chosen? The DU-145 cell line present the highest levels of DYRK2 mRNA, but the K562 and MDA-MB-231 cell lines show a DYRK2 mRNA levels higher than PC-3) are compared with prostate RWPE-1 cell lines. Although in the Methods section it is said that the RWPE-1 cell line is a normal epithelial prostatic cells, it could be explained when fig. 1g is mentioned.

- When the activity of compound 12 and YK-2-69 compound is presented, DYRK1B, DYRK1A, DYRK3 and DYRK4 are mentioned; but these targets were not mentioned before. Some information about these targets would be of interest. Why it is interesting to develop a DYRK2 inhibitor, with selectivity over these targets?

Other minor comments:

- Figure 3d is not the best figure to show that YK-2-69 binds to the ATP-binding pocket of DYRK2.

- Although it is not the main part of the article, some details about the virtual screening and the optimization process of compound 12 to obtain the compound YK-2-69 would be of interest. About the virtual screening, it is described at the Methods section, but more details would be useful. The IC50 values activities of the 15 compounds (from figure S2) initially assayed will be very interesting. About the optimization process of compound 12, only a “systematic structure optimization” is mentioned. Some more information about this optimization and the activities of other compounds

(modified from compound 12) will be of interest. The activities against DYRK2 of all the compounds assayed could be included as supplementary information or submitted to a database like PubChem.

Reviewer #3 (Remarks to the Author):

Overall, this is a comprehensive paper summarizing work from design of a new kinase inhibitor, chemical synthesis to biochemical and biological assays. I can't really judge how relevant another kinase inhibitor is but overall the work is sound and worthy of publication.

However, I also feel that a lot of detail is missing throughout the paper. While the reader is overwhelmed with results in the form of many composite figure not enough care is taken to explain the results and put them into context. There are number of sentence that are simply not clear to the (not-expert) reader, just to give a few examples:

6; Especially, a few known small-molecule inhibitors of DYRK2 are lack of selectivity and druggability and its key interactions with DYRK2

77: ...were further confirmed by crystal structure

677:... Next, protein was prepared by Prepare protein protocol in Discovery Studio (DS) 2020 and then the binding site of 6K0J was defined form PDB site records for further docking

figure caption 1 is unclear and does not explain the figure. Abbreviations colour-codes in 1A,B, and C, are not defined.

Not being an expert in Immunohistochemistry, I don't understand what figure 1E is supposed to show ? Likewise, the cell lines in Figure 1F are not explained, I can only guess that PC-3 and DU-145 are the PC cell lines, clearly, only DU145 shows significantly enhanced mRNA levels. Why do some of the columns have NA on top, others some stars based on a statistical analysis that is not explained. Finally 1G shows only one control cell line, also the figure is very unclear on the last hand top side - supposedly, the key area. I would expect a repeat with more cell lines and the display of the full western blot and not just a very selected area.

The Methods part should describe the cell-culture of all cell lines mentioned here in the results section. Again, there are virtually no references in the methods section.

Figure 2 caption, again leaves a lot of information out. The Western blots (?) again show a very small selected area and are hence difficult to judge. It is not clear to the non-expert reader what figure 1C,D,e, (left hand side) actually show.

The virtual screening procedure should have been explained in a bit more detail in the results section, for example the exclusion criteria such as Lipinski's rule should have been explained here (as this is quite a crude measurement). Also, the choice of the starting library and the property of this library should be explained. Are these random chemicals or enriched as kinase inhibitors ? Which target

functions were used and what were the scores of the top compounds ? And which experiment was used to derive an IC50 value of 263 nm ?

Is the description of interactions from line 160 on based on the docked (optimised ?) structure or the co-crystal structure ? In addition, the methods section lacks all key references. When referring to PDB files the underlying peer-reviewed publication should be cited.

The authors should also explain what the 'systematic structure optimisation' entailed. This step is usually the most challenging one.

Figure 3: The steps from compounds 12 to YK-2-69 should be explained, and the gels should be shown as complete. 3c is pleasing to the eye but it is unclear what actually is shown, What are the scales and what are the color codes . Figure D should be a ribbon diagram and E should show unbiased electron density of the ligand. Once again, the method section lacks all relevant references, including the one for the detector (Pilatus I assume), flash-freezing, structure solution, refinement. As the protein seems to have two independent molecules, I also assume local NCS restraints were used. Molprobity and Phenix references (peer-reviewed publications) must be included. In the crystallographic table B-factors for protein, ligand and waters should be given.

The crystal structure determination is of appropriate quality, however, it is difficult to judge the quality of the ligand position as no unbiased electron density is given. I would have liked to have seen a biophysical method (TSA, ITC or SPR) to give at least an indication of binding or better a proper binding constant.

There are 11 co-crystal structures in the data base, so the authors should compare their binding mode with what had been observed before.

When reporting IC50 values the authors ought to consider significant numbers, for example, and give realistic values.

212 ... against the proliferation of PC-3 and DU-145 cell lines with IC50 values of 1.069 and 211 3.302 μ M, respectively (Fig. 4a).

Figure 4 and 5 again contain a lot of information that is not sufficiently explained in the figure caption. I personally find the tendency to cut out single gel bands from gel somewhat disconcerting.

I am not familiar with animal studies so I can't really comment on those, however, again I noticed that the authors should consider how many significant numbers they quote. It seems unlikely that any of the numbers have precision of 1/10000.

In summary, this seems like an important study but it requires major revisions and additions.

Reviewer #4 (Remarks to the Author):

Summary: In this manuscript, the authors identified DYRK2 as a therapeutic target gene in prostate cancer (PC). By analyzing the TCGA PC dataset, they showed upregulation of DYRK2 in PC compared with matched normal controls, and improved relapse-free survival in the DYRK2-low group. The knockdown of DYRK2 inhibited the tumor growth both in vitro and in vivo. Motivated by these observations, the authors developed a highly selective DYRK2 inhibitor (YK-2-69) with favorable safety properties. They demonstrated the effect of YK-2-69 was similar to that of DYRK2 knockdown based on the transcriptomic analysis on a PC cell line. Furthermore, YK-2-69 showed better anti-tumor activity than a first-line drug in the xenograft mouse model.

This manuscript is well written and nicely demonstrates how anti-cancer drug can be developed by combining multidisciplinary approaches. However, there are a few comments which should be carefully addressed.

Major points:

1. Figure 1B-D: The authors showed that age is associated with the expression of DYRK2 in Figure 1C. I'm wondering whether the observed association in Figure 1B and 1D is still significant after accounting for age.

2. Figure 5: The authors claimed that the effect of YK-2-69 was similar to that of DYRK2 knockdown by comparing differentially expressed genes and enriched signatures from RNA sequencing (RNA-seq) data. However, their main claim is not well supported by data presented. More systematic analysis is required. First, as a negative control, samples treated with the first-line PC drug should be included. Second, for generalizability, one more PC cell line (PC-3?) highly expressing DYRK2 should be analyzed. Third, visualizing transcriptomic differences among samples using principal component analysis or similar methods might be helpful. In addition, using upregulated or downregulated genes in shDYRK2 compared to shVehicle as a signature gene set, the authors can perform GSEA on a ranked gene list of YK-2-69 treated samples compared to WT. Finally, the authors should carefully analyze differentially expressed genes and signatures between shDYRK2 and YK-2-69 treated samples.

Minor points:

1. Line 86-87: "TCGA Cancer Genome Atlas" -> The Cancer Genome Atlas (TCGA)

2. Line 156: "Considering the highly expression level" -> "Considering the high expression level".

3. Figure 3C: What do the color and size of dots mean?

4. Figure 5A and Supplementary Figure 5A: The authors should evaluate the statistical significance of the overlaps.

5. Figure 5B: What does the x-axis mean?

6. Figure 5D and Supplementary Figure 5B: The authors should report P-values.

7. Line 253-255 “Global genome-enrichment analysis showed that DYRK2 KD and YK-2-69 treatment exhibited similar effects on the regulation of a set of functionally important signaling pathways.”: The authors should provide data supporting this sentence.

**** See Nature Research’s author and referees’ website at www.nature.com/authors for information about policies, services and author benefits.**

We would like to express our sincere thanks to the editor and reviewers for the positive and constructive comments! We completely followed the professional suggestions to conduct all the experiments and made a careful revision that helped us to improve our manuscript and to meet the scientific standard of this journal.

Reviewer 1:

In the current paper entitled “targeting dual specificity tyrosine phosphorylation regulated kinase two (DYRK2) with a highly selective inhibitor for the treatment of prostate cancer,” Yuan et al., identified that DYRK2 is expressed in prostate cancer compared to normal tissue and that the genetic knock down of DYRK2 inhibits cancer cell proliferation migration and invasion in vitro and reduces tumor growth in vivo using a subcutaneous xenograft model. Further, the authors identify a novel inhibitor of DYRK2 that has good pharmacokinetics and is efficacious for the treatment of prostate cancer in vivo. There is novelty to paper regarding the identification of DYRK2 and a YK-2-69.

Reply: Thank you very much for the positive comments! This reflects the merits of our manuscript!

Q1. The primary focus is based on DYRK2 expression in prostate cancer tissue vs. normal tissue. What about other family members, DYRK1A, DYRK1B, DYRK3, and DYRK4? Are these expressed in prostate cancer vs. normal tissue and do the other family members correlate with disease progression. There is also never an attempt to validate their findings in human tissues. Does the expression of DYRK correlate with MYC in humans?

A1: Thank you very much for your great advice! Following your professional suggestions, we mined the TCGA database to analyze the expression of DYRK1A, DYRK1B, DYRK3, and DYRK4 in prostate cancer vs. normal tissue and explore the correlation of relapse-free survival with expression of DYRK1A, DYRK1B, DYRK3, and DYRK4. The results were shown in **Supplementary Fig.1**.

The results show that the expression of DYRK1A and DYRK1B are higher in prostate cancer patients when compared with normal controls, while their expression levels do not exhibit any variation among different risk and age groups and neither correlate with relapse-free survival. The expression of DYRK3 and DYRK4 are lower in PCa patients when compared with normal controls, while their expression levels do not exhibit any variation among different risk and age groups and neither correlate with relapse-free survival.

Supplementary Fig.1. Analysis of the expression of DYRK1A, DYRK1B, DYRK3, and DYRK4 in prostate cancer. a-d Comparison of DYRK1A (a), DYRK1B (b), DYRK3 (c), and DYRK4 (d) expression between tumor (red, n = 51) and normal (gray, n = 51) tissues. **e-h** Comparison of DYRK1A (e), DYRK1B (f), DYRK3 (g), and DYRK4 (h) expression between high (red, n = 345) and intermediate (blue, n = 152) risk PCa patients. **i-l** Comparison of DYRK1A (i), DYRK1B (j), DYRK3 (k), and DYRK4 (l) expression between age ≤ 65 (blue, n = 354) and > 65 (yellow, n =

143) PCa patients. **m-p** Kaplan-Meier survival plot of high (red line, n = 246) and low (blue line, n = 246) DYRK1A (**m**), DYRK1B (**n**), DYRK3 (**o**), and DYRK4 (**p**) expression PCa patients. *, $P < 0.05$; **, $P < 0.01$; ***, $P < 0.001$; ****, $P < 0.0001$; ns, not significant.

Furthermore, the correlation of the expression of DYRK with MYC in prostate cancer were also analyzed and shown as below. The results indicate that the expression of DYRK2 is positively correlated with MYC. The expression of DYRK1A, DYRK1B, DYRK3 has no correlation with MYC, while the expression of DYRK4 is negatively correlated with MYC.

Correlation analysis of DYRK expression and MYC expression in prostate cancer. a-e The correlation of DYRK2 (**a**), DYRK1A (**b**), DYRK1B (**c**), DYRK3 (**d**), DYRK4 (**e**) expression with c-MYC expression in prostate cancer were analyzed through mining TCGA database.

Q2. For Figure 1. What is the correlation between DYRK2 and castrate resistant prostate cancer vs. castrate sensitive? The rationale for pursuing DYRK2 is based on 2-cell lines. PC3 represents small cell prostate cancer which is not reflective of adenocarcinoma and is AR negative. Further, DU145 is also androgen negative and derived from a brain metastasis. What is the effect of the drug on more representative prostate cancer cell lines such as 22Rv1/VCAP/C4-2 etc.? This should be demonstrated for robustness of the studies.

A2: Thank you very much for your professional suggestions! We further analyzed the expression of DYRK2 in more prostate cancer cell lines. For the castrate resistant prostate cancer cells PC-3, DU145, and 22Rv1, the expression of DYRK2 was high. While, for castrate sensitive prostate cancer cell LNCaP, the expression of DYRK2 was not high. The results were shown in **Fig. 1g**.

Fig. 1 DYRK2 is highly expressed in PCa. (g) DYRK2 protein levels in different PCa cell lines. Normal prostate epithelial cell RWPE-1 was used as the control.

Besides DU145 cells, we followed your professional suggestion to further evaluate the effects of the drug YK-2-69 on PC-3 and 22Rv1 cells. YK-2-69 inhibited cell growth and metastasis and induced apoptosis in PCa DU145, PC-3 and 22Rv1 cells. These results were shown in **Fig. 5** and **Supplementary Fig. 7**.

Fig. 5 YK-2-69 inhibited cell growth and metastasis and induced apoptosis in PCa cells. **a** Antiproliferative activity of YK-2-69 against DU145 and 22Rv1 shDYRK2 cells. **b, c** Effects of YK-2-69 (4, 8, 12 μM or 0.5, 1, 2 μM), palbociclib (2 or 0.5 μM), or enzalutamide (40 or 20 μM) treatment on the DU145 (**b**) and 22Rv1 (**c**) cells viability during a 5-day course. **d, e** Quantification of DU145 (**d**) and 22Rv1 (**e**) cells colony numbers. Before being plated on the 24-well plate for colony formation, DU145 and 22Rv1 cells were treated with DMSO, YK-2-69 (2, 4, 8 or 0.5, 1, 2

μM), palbociclib (2 or 0.5 μM), or enzalutamide (40 or 20 μM) for 48 h. **f, g** Quantification of migration ability of DU145 (**f**) and 22Rv1(**g**) cells after treatment with DMSO, YK-2-69 (2, 4, 8 μM or 0.5, 1, 2 μM), palbociclib (2 or 0.5 μM), or enzalutamide (40 or 20 μM) for 48 h. **h, i** Quantification of invasion ability of DU145 (**h**) and 22Rv1(**i**) cells after treatment with DMSO, YK-2-69 (2, 4, 8 or 0.5, 1, 2 μM), palbociclib (2 or 0.5 μM), or enzalutamide (40 or 20 μM) for 48 h. **j, k** Apoptosis of DU145 (**j**) and 22Rv1 (**k**) cells after treatment with DMSO, YK-2-69 (2, 4, 8 or 0.5, 1, 2 μM), palbociclib (2 or 0.5 μM), or enzalutamide (40 or 20 μM) for 48 h determined by flow cytometry. **l, m** Cell cycle phase distribution of DU145 (**l**) and 22Rv1 (**m**) cells after treatment with DMSO, YK-2-69 (2, 4, 8 or 0.5, 1, 2 μM), palbociclib (2 or 0.5 μM), or enzalutamide (40 or 20 μM) for 48 h determined by flow cytometry.

*, $P < 0.05$; **, $P < 0.01$; ***, $P < 0.001$; ****, $P < 0.0001$; Unpaired two-tailed Student's t test. Error bar, mean \pm SD, $n = 3$.

Supplementary Fig. 7 YK-2-69 suppressed cell proliferation and metastasis and promoted apoptosis in PCa cells.

a-c Antiproliferative activities of YK-2-69 (**a**), palbociclib (**b**) and enzalutamide (**c**) against of DU145, PC-3, 22RV1, and RWPE-1 cell lines. **d** Effects of YK-2-69 (1, 2, 4 μM), palbociclib (4 μM), or enzalutamide (40 μM) treatment on the viability of the PC-3 cells during a 5-day course. **e**

Quantification of PC-3 cells colony numbers. Before being plated on the 24-well plate for colony formation, PC-3 cells were treated with DMSO, YK-2-69(1, 2, 4 μ M), palbociclib (2 μ M), or enzalutamide (40 μ M) for 48 h. **f, g** Quantification of migration (**f**) and invasion (**g**) ability of PC-3 after treatment with DMSO, YK-2-69(1, 2, 4 μ M), palbociclib (2 μ M), or enzalutamide (40) for 48 h. **h** Apoptosis of PC-3 cells after treatment with DMSO, YK-2-69 (1, 2, 4 μ M), palbociclib (2 μ M), or enzalutamide (40 μ M) for 48 h determined by flow cytometry. **i** Cell cycle phase distribution of PC-3 cells after treatment with DMSO, YK-2-69 (1, 2, 4 μ M), palbociclib (2 μ M), or enzalutamide (40 μ M) for 48 h determined by flow cytometry. **j** Western blotting analysis of indicated proteins in DU145 cells after treatment with DMSO or YK-2-69 (2, 4, 8 μ M) for 48 h.

*, $P < 0.05$; **, $P < 0.01$; ***, $P < 0.001$; ****, $P < 0.0001$; ns, not significant; Unpaired two-tailed Student's t test. Error bar, mean \pm SD, $n = 3$.

Q3. Figure 2. Invasion and migration assay results may be confounded by the effects of DYRK2 knockdown on growth – less migration/invasion could be due to slower growth over 24 hours. The growth rate of the DYRK2 knockdown cells is remarkable given that DYRK2 is knocked down approx. by 75%. It looks as though they did not grow at all subcutaneously raising the questions as to what is being analyzed in Fig. 2H. What is the efficacy of DYRK2 knockdown on other prostate cancer cell lines *in vivo*?

A3: Thank you very much for the great question! The effects of DYRK2 knockdown on growth of DU145 and 22Rv1 cells were evaluated, and the results demonstrated that the knockdown of DYRK2 had no effects on the growth of DU145 and 22Rv1 cell lines over 24 hours. The results were shown in **Fig. 2c** and **2d**.

Fig. 2 Knock-down of DYRK2 inhibited PCa *in vitro* and *in vivo*. (c, d) Cell viability of DU145 shNC/shDYRK2 cells (c) and 22Rv1 shNC/shDYRK2 cells (d) during a 5-day course.

As shown in the original mice and tumor tissues pictures of DU145 shDYRK2 group, the tumor lumps of DU145 shDYRK2 mice group grew very slowly, but did form.

DU145 shDYRK2 group

DU145 shDYRK2 group

Besides DU145 cells, the efficacy of DYRK2 knockdown on 22Rv1 cells *in vivo* was also evaluated. The body weight of mice increased normally but no visible tumors were detected in the 22Rv1 shDYRK2 group, which was shown in **Supplementary Fig. 2h, i**. The original mice pictures of 22Rv1 shNC and shDYRK2 group were also shown as below.

Supplementary Fig.2 DYRK2 KD significantly inhibited PCa. (h, i) BALB/c mice were implanted subcutaneously with 22Rv1 shNC (n = 6) and shDYRK2 (n = 8) cells. Tumor volume of mice (**h**) and Body weight of mice (**i**) were measured every two days. The mice were euthanized at 19th day.

Q4. Figure 4 – Need to show the effect of the YK-2-69 on multiple cell lines including RWPE-1. Specificity of YK-2-69 against other PCa cell lines in which DYRK2 is knocked down should be demonstrated. Further the effects of YK-2-69 on other cell lines should be demonstrated on additional PCa cell lines with regard to growth/migration.

A4: Thank you very much for your great advice! We followed your professional suggestion to evaluate the antiproliferative activity of YK-2-69 on PC-3, DU145, 22Rv1, and RWPE-1 cells. YK-2-69 exhibited potent inhibitory activity against PC-3, DU145, and 22Rv1 cell lines, while displayed weak inhibitory activity against RWPE-1 cells. The results were shown in **Supplementary Fig. 7a-c**.

For shDYRK2 22Rv1 cells, YK-2-69 exhibited no inhibitory activity on the proliferation with an IC_{50} value of more than 80 μ M. The results were shown in **Fig. 5a**.

Besides DU145 cells, YK-2-69 also demonstrated potent inhibition on the cell growth and migration for PC-3 and 22Rv1 cells. The results were shown in **Fig. 5** and **Supplementary Fig. 7**.

Fig. 5 YK-2-69 inhibited cell growth and metastasis and induced apoptosis in PCa cells. **a** Antiproliferative activity of YK-2-69 against DU145 and 22Rv1 shDYRK2 cells. **b, c** Effects of YK-2-69 (4, 8, 12 μM or 0.5, 1, 2 μM), palbociclib (2 or 0.5 μM), or enzalutamide (40 or 20 μM) treatment on the DU145 (**b**) and 22Rv1 (**c**) cells viability during a 5-day course. **d, e** Quantification of DU145 (**d**) and 22Rv1 (**e**) cells colony numbers. Before being plated on the 24-well plate for colony formation, DU145 and 22Rv1 cells were treated with DMSO, YK-2-69 (2, 4, 8 or 0.5, 1, 2

μM), palbociclib (2 or 0.5 μM), or enzalutamide (40 or 20 μM) for 48 h. **f, g** Quantification of migration ability of DU145 (**f**) and 22Rv1(**g**) cells after treatment with DMSO, YK-2-69 (2, 4, 8 μM or 0.5, 1, 2 μM), palbociclib (2 or 0.5 μM), or enzalutamide (40 or 20 μM) for 48 h. **h, i** Quantification of invasion ability of DU145 (**h**) and 22Rv1(**i**) cells after treatment with DMSO, YK-2-69 (2, 4, 8 or 0.5, 1, 2 μM), palbociclib (2 or 0.5 μM), or enzalutamide (40 or 20 μM) for 48 h. **j, k** Apoptosis of DU145 (**j**) and 22Rv1 (**k**) cells after treatment with DMSO, YK-2-69 (2, 4, 8 or 0.5, 1, 2 μM), palbociclib (2 or 0.5 μM), or enzalutamide (40 or 20 μM) for 48 h determined by flow cytometry. **l, m** Cell cycle phase distribution of DU145 (**l**) and 22Rv1 (**m**) cells after treatment with DMSO, YK-2-69 (2, 4, 8 or 0.5, 1, 2 μM), palbociclib (2 or 0.5 μM), or enzalutamide (40 or 20 μM) for 48 h determined by flow cytometry.

*, $P < 0.05$; **, $P < 0.01$; ***, $P < 0.001$; ****, $P < 0.0001$; Unpaired two-tailed Student's t test. Error bar, mean \pm SD, $n = 3$.

Supplementary Fig. 7 YK-2-69 suppressed cell proliferation and metastasis and promoted apoptosis in Pca cells. a-c Antiproliferative activities of YK-2-69 (a), palbociclib (b) and enzalutamide (c) against of DU145, PC-3, 22Rv1, and RWPE-1 cell lines. d Effects of YK-2-69 (1, 2, 4 μM), palbociclib (4 μM), or enzalutamide (40 μM) treatment on the viability of the PC-3 cells during a 5-day course. e Quantification of PC-3 cells colony numbers. Before being plated on the

24-well plate for colony formation, PC-3 cells were treated with DMSO, YK-2-69 (1, 2, 4 μ M), palbociclib (2 μ M), or enzalutamide (40 μ M) for 48 h. **f, g** Quantification of migration (**f**) and invasion (**g**) ability of PC-3 after treatment with DMSO, YK-2-69(1, 2, 4 μ M), palbociclib (2 μ M), or enzalutamide (40) for 48 h. **h** Apoptosis of PC-3 cells after treatment with DMSO, YK-2-69 (1, 2, 4 μ M), palbociclib (2 μ M), or enzalutamide (40 μ M) for 48 h determined by flow cytometry. **i** Cell cycle phase distribution of PC-3 cells after treatment with DMSO, YK-2-69 (1, 2, 4 μ M), palbociclib (2 μ M), or enzalutamide (40 μ M) for 48 h determined by flow cytometry. **j** Western blotting analysis of indicated proteins in DU145 cells after treatment with DMSO or YK-2-69 (2, 4, 8 μ M) for 48 h. *, $P < 0.05$; **, $P < 0.01$; ***, $P < 0.001$; ****, $P < 0.0001$; ns, not significant; Unpaired two-tailed Student's t test. Error bar, mean \pm SD, $n = 3$.

Q5. Figure 5 is also solely based on DU145 with no validation of findings in other cell line models.

A5: Thank you very much for your great advice! Besides DU145 cells, we followed your professional suggestion to validate the findings in 22Rv1 cells.

The results were shown in **Supplementary Fig. 8c**. Both DYRK2 KD and YK-2-69 treatment induced significant inhibition of MYC targets V1 and E2F targets in 22Rv1 cells.

Supplementary Fig. 8 Transcriptome-wide RNA sequencing assays in PCa 22Rv1 cells. (c) The signaling pathways enriched in different groups obtained through Gene set enrichment analysis.

A6. Fig. 6, remarkably, enzalutamide looks to have a significant effect on the AR negative DU145 cell line in vivo again raising concerns as to the robustness and rigor of the findings.

A6: Thank you very much! We agreed that enzalutamide was not effective for hormone independent prostate cancer patients, while no paper was found to demonstrate that enzalutamide had no effects on DU145 xenograft model. Each small molecule inhibitor has more than one target, and enzalutamide could have effects on other targets to show potent antitumor activity in the DU145 xenograft model.

To prove the authenticity of our experiments, we provided the original picture of tumor tissues in the manuscript (**Fig. 7d**). Furthermore, we will research the mechanism that enzalutamide displays potent antitumor activity for the DU145 xenograft model in the future.

Fig. 7. YK-2-69 demonstrated remarkable antitumor activities in vivo. (d) BALBc nude mice received subcutaneous injection of 1×10^7 DU145 cells in the right flank. When tumors grew about 80-100 mm³, mice (n = 10/group) were orally administered vehicle, palbociclib (100 mg/kg), enzalutamide (100 mg/kg) and YK-2-69 (100 and 200 mg/kg) every day. After 35 days, mice in control group were sacrificed. After 49 days, mice of treatment groups were sacrificed. Tumor tissues of each group were weighed and photographed.

Minor Points

Q1. Enzalutamide and Palbociclib should be used as a control of *in vitro* data, especially Fig 4.

A1. Thank you very much! We followed your great suggestions to use enzalutamide and palbociclib as a control of *in vitro* data. The results were shown in **Fig. 5** and **Supplementary Fig. 7**.

Fig. 5 YK-2-69 inhibited cell growth and metastasis and induced apoptosis in Pca cells. a

Antiproliferative activity of YK-2-69 against DU145 and 22Rv1 shDYRK2 cells. **b, c** Effects of YK-2-69 (4, 8, 12 μ M or 0.5, 1, 2 μ M), palbociclib (2 or 0.5 μ M), or enzalutamide (40 or 20 μ M) treatment on the DU145 (**b**) and 22Rv1 (**c**) cells viability during a 5-day course. **d, e** Quantification of DU145 (**d**) and 22Rv1(**e**) cells colony numbers. Before being plated on the 24-well plate for colony formation, DU145 and 22Rv1 cells were treated with DMSO, YK-2-69 (2, 4, 8 or 0.5, 1, 2 μ M), palbociclib (2 or 0.5 μ M), or enzalutamide (40 or 20 μ M) for 48 h. **f, g** Quantification of migration ability of DU145 (**f**) and 22Rv1(**g**) cells after treatment with DMSO, YK-2-69 (2, 4, 8 μ M or 0.5, 1, 2 μ M), palbociclib (2 or 0.5 μ M), or enzalutamide (40 or 20 μ M) for 48 h. **h, i** Quantification of invasion ability of DU145 (**h**) and 22Rv1(**i**) cells after treatment with DMSO, YK-2-69 (2, 4, 8 or 0.5, 1, 2 μ M), palbociclib (2 or 0.5 μ M), or enzalutamide (40 or 20 μ M) for 48 h. **j, k** Apoptosis of DU145 (**j**) and 22Rv1 (**k**) cells after treatment with DMSO, YK-2-69 (2, 4, 8 or 0.5, 1, 2 μ M), palbociclib (2 or 0.5 μ M), or enzalutamide (40 or 20 μ M) for 48 h determined by flow cytometry. **l, m** Cell cycle phase distribution of DU145 (**l**) and 22Rv1 (**m**) cells after treatment with DMSO, YK-2-69 (2, 4, 8 or 0.5, 1, 2 μ M), palbociclib (2 or 0.5 μ M), or enzalutamide (40 or 20 μ M) for 48 h determined by flow cytometry.

*, $P < 0.05$; **, $P < 0.01$; ***, $P < 0.001$; ****, $P < 0.0001$; Unpaired two-tailed Student's t test.

Error bar, mean \pm SD, $n = 3$.

Supplementary Fig. 7 YK-2-69 suppressed cell proliferation and metastasis and promoted apoptosis in Pca cells. a-c Antiproliferative activities of YK-2-69 (a), palbociclib (b) and enzalutamide (c) against of DU145, PC-3, 22Rv1, and RWPE-1 cell lines. d Effects of YK-2-69 (1, 2, 4 μM), palbociclib (4 μM), or enzalutamide (40 μM) treatment on the viability of the PC-3 cells during a 5-day course. e Quantification of PC-3 cells colony numbers. Before being plated on the

24-well plate for colony formation, PC-3 cells were treated with DMSO, YK-2-69 (1, 2, 4 μ M), palbociclib (2 μ M), or enzalutamide (40 μ M) for 48 h. **f, g** Quantification of migration (**f**) and invasion (**g**) ability of PC-3 after treatment with DMSO, YK-2-69(1, 2, 4 μ M), palbociclib (2 μ M), or enzalutamide (40) for 48 h. **h** Apoptosis of PC-3 cells after treatment with DMSO, YK-2-69 (1, 2, 4 μ M), palbociclib (2 μ M), or enzalutamide (40 μ M) for 48 h determined by flow cytometry. **i** Cell cycle phase distribution of PC-3 cells after treatment with DMSO, YK-2-69 (1, 2, 4 μ M), palbociclib (2 μ M), or enzalutamide (40 μ M) for 48 h determined by flow cytometry. **j** Western blotting analysis of indicated proteins in DU145 cells after treatment with DMSO or YK-2-69 (2, 4, 8 μ M) for 48 h. *, $P < 0.05$; **, $P < 0.01$; ***, $P < 0.001$; ****, $P < 0.0001$; ns, not significant; Unpaired two-tailed Student's t test. Error bar, mean \pm SD, $n = 3$.

Q2. Some typos throughout. Line 73 “inhibits metastasis *in vitro*” Line 117 “inhibited cell cloning”

A2. Thank you very much! We have revised these typos and double check all the manuscript.

Such as, “Knock-down of DYRK2 in PCa greatly promotes apoptosis, causes a G1 arrest of cell cycle and inhibits metastasis *in vitro*” was revised to “Knock-down of DYRK2 in PCa cells greatly suppressed cell proliferation and metastasis, promoted apoptosis, and caused a G1 arrest of the cell cycle.”

“The knock-down of DYRK2 in DU145 cells significantly inhibited cell cloning, migration, and invasion” was revised to “The knock-down of DYRK2 in PCa cells significantly inhibited cell growth and metastasis.”

Q3. Statistics should be included for the *in vivo* studies.

A3. Thank you very much! We followed your great suggestions to add statistics for the *in vivo* studies. The results were shown in **Fig. 7** and **Supplementary Fig. 10**.

Fig. 7. YK-2-69 demonstrated remarkable antitumor activities *in vivo*. (a-d) BABLc nude mice received subcutaneous injection of 1×10^7 DU145 cells in the right flank. When tumors grew about 80-100 mm³, mice (n = 10/group) were orally administered vehicle, palbociclib (100 mg/kg), enzalutamide (100 mg/kg) and YK-2-69 (100 and 200 mg/kg) every day. Tumor volumes (a) and body weight of mice (b) were measured every two days. After 35 days, mice in control group were sacrificed. After 49 days, mice of treatment groups were sacrificed. Tumor issues of each group were weighed (c) and then photographed (d).

Supplementary Fig.10 YK-2-69 significantly inhibited tumor growth *in vivo*. BALBc nude mice received subcutaneous injection of 1×10^7 PC-3 cells in the right flank. When tumors grew about 80-100 mm³, four group mice (n = 8/group) were orally administrated vehicle, palbociclib (100 mg/kg), and YK-2-69 (100 and 200 mg/kg) every day, respectively. Tumor volumes (a) and body weight of mice (b) were measured every two days. After 29 days, mice were sacrificed. Tumor issues of each group were weighed (c) and then photographed (d).

Reviewer 2:

The manuscript identifies DYRK2 as a novel drug target for prostate cancer and discover a highly selective DYRK2 inhibitor with favorable druggability for the treatment of prostate cancer (PCa). The manuscript is very complete as includes the demonstrations that DYRK2 is a potential therapeutic target for PCa, the identification and improvement of a highly selective inhibitor of DYRK2, the cocrystal structure of this inhibitor with DYRK2, the in vitro and in vivo activities of this compound and its pharmacokinetic parameters in rats. This manuscript could allow the development of new treatments for prostate cancer. The manuscript is well written.

Reply: Thank you very much for the positive comments! This reflects the merits of our manuscript! We completely followed your professional suggestions to conduct all the experiments and made a careful revision that helped us to improve our manuscript and to meet the scientific standard of this journal.

Q1. Figure 1 describes the higher expression of DYRK2 in PCa patients, the differences of expression of DYRK2 between patients with high or intermediate risk, differences in relapse-free survival, In this part, the mRNA level of DYRK2 were evaluated in 11 different cell lines, showing that PC-3 and DU145 PCa cell lines showed much higher DYRK2 mRNA levels (Fig. 1f). But, which was the objective of analyzing these 11 different cell lines? All these are PCa cell lines? The addition of a sentence explaining a little more the main purpose of this experiment would be useful. In this figure, the *, ** and *** symbols represent the unpaired Student's t-test $p < 0.05$, $p < 0.01$ and $p < 0.001$, respectively; but which is the reference group? What does NA means?

A1. Thank you very much for your professional suggestions! The analysis of these 11 different cell lines was the preliminary exploration of the DYRK2 expression in different cancer lines, and HEK-293 was used as the reference group. NA should be NS (not significant).

To focus on prostate cancer, the figure about the mRNA level of DYRK2 in 11 different cell lines was replaced with more prostate cancer cell lines 22Rv1 and LNCaP, which were used to determine the DYRK2 expression. The results were shown in **Fig. 1g**.

Fig. 1 DYRK2 is highly expressed in PCa. (g) DYRK2 protein levels in different PCa cell lines. Normal prostate epithelial cell RWPE-1 was used as the control.

Q2. In addition, figures 1a, 1b and 1c represent the values of DYRK2 expression measured in FPKM, but this unit is not defined.

A2. Thank you very much! We followed your great suggestions to define the unit of FRKM (fragment per killo million).

Fig. 1 DYRK2 is highly expressed in PCa. a-c Comparison of DYRK2 expression between tumor (red, n = 51) and normal (gray, n = 51) tissues (a), high (red, n = 345) and intermediate (blue, n = 152) risk PCa patients (b), age <= 65 (blue, n = 354) or > 65 (yellow, n = 143) PCa patients (c) in TCGA database. FRKM: fragment per killo million.

Q3. Also in figure 1g the protein levels of DYRK2 in PC-3 and DU145 cell lines (why these two cell lines were chosen? The DU145 cell line present the highest levels of DYRK2 mRNA, but the K562 and MDA-MB-231 cell lines show a DYRK2 mRNA levels higher than PC-3) are compared with prostate RWPE-1 cell lines. Although in the Methods section it is said that the RWPE-1 cell line is a normal epithelial prostatic cells, it could be explained when fig. 1g is mentioned.

A3. Thank you very much for your great suggestions! The analysis of these 11 different cell lines

was the preliminary exploration of the DYRK2 expression in different cancer lines. Prostate cancer was the focus of our study, and K562 and MDA-MB-231 cell lines were not the focus. Therefore, PC-3 and DU145 cell lines were chosen. Furthermore, more prostate cancer cell lines 22Rv1 and LNCaP were used to determine the DYRK2 expression.

Also, we followed your professional advice to explain that the normal epithelial prostatic cells RWPE-1 was used as the control when fig. 1g is mentioned.

Fig. 1 DYRK2 is highly expressed in PCa. (g) DYRK2 protein levels in different PCa cell lines. Normal prostate epithelial cell RWPE-1 was used as the control.

Q4. When the activity of compound **12** and YK-2-69 compound is presented, DYRK1A, DYRK1B, DYRK3 and DYRK4 are mentioned; but these targets were not mentioned before. Some information about these targets would be of interest. Why it is interesting to develop a DYRK2 inhibitor, with selectivity over these targets?

A4. Thank you very much for your professional suggestions! We followed your great advice to provide more information about DYRK1A, DYRK1B, DYRK3 and DYRK4 in the introduction part. Furthermore, the expression of DYRK1A, DYRK1B, DYRK3, and DYRK4 in prostate cancer vs. normal tissue and their correlation with disease progression were analyzed through mining TCGA database. The results were shown in **Supplementary Fig. 1**.

The results showed that the expression of DYRK1A and DYRK1B are higher in prostate cancer patients when compared with normal controls, while their expression levels do not exhibit any variation among different risk and age groups and neither correlate with relapse-free survival. The expression of DYRK3 and DYRK4 are lower in PCa patients when compared with normal controls, while their expression levels do not exhibit any variation among different risk and age groups and neither correlate with relapse-free survival.

All the results indicated that DYRK2 was the potential target for PCa treatment. While, other DYRK family members, DYRK1A, DYRK1B, DYRK3, and DYRK4 were not great candidate targets for anti-PCa drugs. Therefore, developing a highly selective DYRK2 inhibitor is an interesting and good strategy for PCa treatment.

Supplementary Fig.1. Analysis of the expression of DYRK1A, DYRK1B, DYRK3, and DYRK4 in prostate cancer. a-d Comparison of DYRK1A (a), DYRK1B (b), DYRK3 (c), and DYRK4 (d) expression between tumor (red, n = 51) and normal (gray, n = 51) tissues. **e-h** Comparison of DYRK1A (e), DYRK1B (f), DYRK3 (g), and DYRK4 (h) expression between high (red, n = 345)

and intermediate (blue, $n = 152$) risk PCa patients. **i-l** Comparison of DYRK1A (**i**), DYRK1B (**j**), DYRK3 (**k**), and DYRK4 (**l**) expression between age ≤ 65 (blue, $n = 354$) and > 65 (yellow, $n = 143$) PCa patients. **m-p** Kaplan-Meier survival plot of high (red line, $n = 246$) and low (blue line, $n = 246$) DYRK1A (**m**), DYRK1B (**n**), DYRK3 (**o**), and DYRK4 (**p**) expression PCa patients. *, $P < 0.05$; **, $P < 0.01$; ***, $P < 0.001$; ****, $P < 0.0001$; ns, not significant.

Other minor comments:

Q1. Figure 3d is not the best figure to show that YK-2-69 binds to the ATP-binding pocket of DYRK2.

A1. Thank you very much! We followed your great suggestions to provide more figures to display that YK-2-69 binds to the ATP-binding pocket of DYRK2. The results were shown in **Fig. 4**.

Fig. 4 Co-crystal structure of YK-2-69 with DYRK2. **a** the F_0-F_C omitted map is contoured at 3.0σ and shown as a blue mesh, which reveals the presence of YK-2-69. DYRK2 is shown as ribbons. PDB ID 7EJV. **b** YK-2-69 occupies the ATP binding pocket of DYRK2. DYRK2 is shown as surface, and YK-2-69 is shown as spheres. **c** Detailed interactions between YK-2-69 and DYRK2 in

co-crystal structure.

Q2. Although it is not the main part of the article, some details about the virtual screening and the optimization process of compound **12** to obtain the compound YK-2-69 would be of interest. About the virtual screening, it is described at the Methods section, but more details would be useful.

A2. Thank you very much! We followed your great suggestions to provide more details about the virtual screening in the Supplementary Experimental Procedures and added the optimization process of compound **12** to obtain compound YK-2-69 in the Methods section.

“Structure-based virtual screening. Specs database (<http://www.specs.net>) contains more than 200,000 single synthesized, well-characterized and drug-like small molecules, and they are easy to obtain. It is a great choice for us to find a lead compound. Meanwhile, an in-house compound library has been established, which is mainly made up of kinase inhibitors. Therefore, the Specs database (221,097 compounds) and in-house compound library (about 3,000 compounds) were combined for the virtual screening. To eliminate compounds with poor drug-like properties, the combined ligand database was employed into DS 2020 and then filtered based on ‘Lipinski Rule of Five’¹, ‘Veber Rule’², and ‘Pan Assay Interference Compounds (PAINS)’^{3,4}. ‘Lipinski Rule of Five’ is used to reserve drug-like ligands which have the following features: no more than 5 hydrogen bond donors, no more than 10 hydrogen bond acceptors, molecular weight no more than 500, and LogP no more than 5. The parameter “Number of Violations Allowed” in the ‘Lipinski Rule of Five’ is 1, which means that the ligands with at least three features mentioned can be reserved. ‘Veber Rule’ is used to select drug candidates with good oral bioavailability, which have the following features: no more than 10 rotatable bonds, polar surface area no more than 1,400 Å², or no more than 12 hydrogen bond donors and acceptors. ‘PAINS’ is used to remove ligands that are not drug-like or lead-like. After the elimination of ligands with poor drug-like properties, 195,483 compounds were reserved, which were then prepared through the *Prepare Ligands* protocol of DS 2020 to add hydrogens, remove duplicates, apply the CHARMM force field and minimize. The prepared ligands can be used for the following virtual screening.

The co-crystal structure of DYRK2 (PDB ID: 6K0J) was downloaded from RCSB Protein Data Bank⁵, and the disordered conformations, crystal water, and ligand in the co-crystal were removed. The protein was applied to the CHARMM force field and prepared via *Prepare Protein* protocol of

DS 2020. The binding site was defined by *From PDB Site Records* protocol for further molecule docking.

The structure-based virtual screening procedure contains two programs, LibDock and CDOCKER. LibDock is a molecular docking program based on a high-throughput algorithm, which matches the poses of the ligand according to the structural feature of the receptor⁶. CDOCKER program employs high-temperature molecular dynamics to generate conformation and random rotation of the ligand⁷. Compared with Libdock, CDOCKER is more precise, but it needs more time. Therefore, Libdock was firstly used for primary screening, then CDOCKER was used for further screening.

195,483 prepared compounds firstly docked with DYRK2 protein via the Libdock protocol of DS2020. The pose with the highest score of each ligand was picked out, and ligands ranked based on the 'Libdock Score'. Then 9,696 ligands with a LibDock score of more than 125 were remained, which were further filtered through CDOCKER. Conformation of each ligand was reserved in lowest energy and ligands ranked according to '-CDOCKER INTERACTION ENERGY'. Finally, 2,724 ligands with a "- CDOCKER INTERACTION ENERGY" of more than 55 remained.

Find Diverse Molecules protocol is used to find a diverse subset of ligands from the input ligands. Therefore, to further select ligands from 2,724 ligands, these ligands were clustered into 100 clusters via *Find Diverse Molecules* protocol based on the functional-class fingerprint of diameter 6 (FCFP_6) properties. From 100 clusters, 15 compounds (Supplementary Fig. 3) were selected through visual inspection. In the end, the DYRK2 inhibitory activity of these 15 compounds was evaluated. Among these 15 compounds, compound **12** exhibited the best inhibitory activity against DYRK2 with an IC₅₀ value of 263 nM, and IC₅₀ values of the other 14 compounds were all more than 10 μM."

Structural optimization. To obtain more potent DYRK2 inhibitor, structural optimization on the lead compound **12** was conducted (Supplementary Fig. 4). Firstly, the isopropyl group on the piperazine was replaced with different sizes substituents (**16-18**), among which compounds **16** and **18** with a small ethyl or hydrogen group showed similar activity as lead **12**, while compound **17** with a big Boc group exhibited significantly decreased activity. When benzothiazole was changed to pyrazolo[1,5-a]pyrimidine (**19**), DYRK2 inhibitory activity significantly decreased. Therefore, benzothiazole was retained in the following structural optimization. To our delight, the optimization on the linker between pyridine and piperazine (**20**: 35 nM and **21**: 85 nM) remarkably improved

DYRK2 inhibitory activity, especially the carbonyl linker. Subsequently, the change of pyridine to pyrimidine (**22**: 697 nM) also decreased DYRK2 inhibitory activity. Therefore, pyridine is a favorable moiety for the maintenance of activity. After confirming the benzothiazole, pyridine, and carbonyl linker as the optimized groups, the subsequent modification was carried out by introducing substituents on the benzothiazole. The introduction of methyl substituent generated **23** and **24**, which exhibited stronger inhibition on DYRK2 with IC₅₀ values of 22 and 27 nM, respectively. The introduction of dimethylamine substituent (**25**, **26**) further improved DYRK2 inhibitory activity, and **26** exhibited the most potent inhibition on DYRK2 with an IC₅₀ value of 9 nM. The change of pyrimidine to thieno[3,2-d]pyrimidine generated **27**, which showed no inhibition on DYRK2. Taken together, **26** was confirmed as the most potent DYRK2 inhibitor, which was re-named as YK-2-69 and selected for further biological evaluation.”

Q3. The IC₅₀ values activities of the 15 compounds (from figure S2) initially assayed will be very interesting.

A3. Thank you very much! The IC₅₀ values activities of the 15 compounds (from figure S2) were provided in the manuscript. We also provided more information, including molecular weight, ALogP, number of hydrogen bond donors and acceptors in Lipinski rule in **Supplementary Table 1**.

“**Supplementary Table 1: Virtual screening and biological assay of 15 candidate hits^a.**”

No	Score 1	Score 2	MW	LogP	H Donors/ Acceptors	IC ₅₀ (μM)
1	127.32	56.92	478.64	7.59	0/2	> 10
2	140.06	56.22	494.62	7.72	1/3	> 10
3	131.23	61.67	480.99	7.45	1/5	> 10
4	141.63	55.19	580.73	4.26	0/10	> 10
5	132.21	57.18	478.54	5.21	1/8	> 10
6	135.98	56.50	520.62	4.52	0/8	> 10
7	146.32	57.95	561.68	4.60	1/9	> 10
8	131.01	57.47	488.29	2.93	0/10	> 10
9	126.54	55.25	487.61	5.99	3/6	> 10
10	136.95	61.37	499.31	3.04	1/10	> 10
11	130.86	55.06	498.92	4.63	1/8	> 10
12	140.30	63.44	463.58	4.50	1/7	0.263
13	128.06	58.36	474.60	3.86	2/8	> 10

14	125.14	55.49	465.59	3.16	1/8	> 10
15	125.94	55.69	458.49	3.42	1/10	> 10
Staurosporine						0.179

“Score 1: Libdock Score; Score 2: -CDOCKER INTERACTION ENERGY; MW: Molecular Weight; H Donors/ Acceptors: Number of hydrogen bond donors and acceptors in Lipinski rule; Staurosporine is the positive control in DYRK2 kinase assays.”

Q4. About the optimization process of compound **12**, only a “systematic structure optimization” is mentioned. Some more information about this optimization and the activities of other compounds (modified from compound **12**) will be of interest. The activities against DYRK2 of all the compounds assayed could be included as supplementary information or submitted to a database like PubChem.

A4. Thank you very much for your professional suggestions! We provided all the details about the optimization process of compound **12** to obtain compound YK-2-69 in the Methods section. The activities against DYRK2 of all these compounds were included in supplementary information.

“**Structural optimization.** To obtain more potent DYRK2 inhibitor, structural optimization on the lead compound **12** was conducted (Supplementary Fig. 4). Firstly, the isopropyl group on the piperazine was replaced with different sizes substituents (**16-18**), among which compounds **16** and **18** with a small ethyl or hydrogen group showed similar activity as lead **12**, while compound **17** with a big Boc group exhibited significantly decreased activity. When benzothiazole was changed to pyrazolo[1,5-a]pyrimidine (**19**), DYRK2 inhibitory activity significantly decreased. Therefore, benzothiazole was retained in the following structural optimization. To our delight, the optimization on the linker between pyridine and piperazine (**20**: 35 nM and **21**: 85 nM) remarkably improved DYRK2 inhibitory activity, especially the carbonyl linker. Subsequently, the change of pyridine to pyrimidine (**22**: 697 nM) also decreased DYRK2 inhibitory activity. Therefore, pyridine is a favorable moiety for the maintenance of activity. After confirming the benzothiazole, pyridine, and carbonyl linker as the optimized groups, the subsequent modification was carried out by introducing substituents on the benzothiazole. The introduction of methyl substituent generated **23** and **24**, which exhibited stronger inhibition on DYRK2 with IC₅₀ values of 22 and 27 nM, respectively. The

introduction of dimethylamine substituent (**25**, **26**) further improved DYRK2 inhibitory activity, and **26** exhibited the most potent inhibition on DYRK2 with an IC₅₀ value of 9 nM. The change of pyrimidine to thieno[3,2-d]pyrimidine generated **27**, which showed no inhibition on DYRK2. Taken together, **26** was confirmed as the most potent DYRK2 inhibitor, which was re-named as YK-2-69 and selected for further biological evaluation.”

Supplementary Fig. 4 Chemical structures and DYRK2 kinase IC₅₀ values of compounds 16-27.

To further improve DYRK2 inhibitory activity, representative derivatives **16-27** were synthesized and further tested their inhibitory activity against DYRK2 kinase. Among them, compound **26** (re-named as YK-2-69) exhibited the most potent DYRK2 inhibitory activity with an IC₅₀ value of 9 nM.”

Reviewer 3:

Overall, this is a comprehensive paper summarizing work from design of a new kinase inhibitor, chemical synthesis to biochemical and biological assays. I can't really judge how relevant another kinase inhibitor is but overall the work is sound and worthy of publication.

Reply: Thank you very much for the positive comments! This reflects the merits of our manuscript! We completely followed your professional suggestions to conduct all the experiments and made a careful revision that helped us to improve our manuscript and to meet the scientific standard of this journal.

Q1. There are number of sentences that are simply not clear to the (not-expert) reader, just to give a few examples:

- a. Especially, a few known small-molecule inhibitors of DYRK2 are lack of selectivity and druggability
- b. and its key interactions with DYRK2 were further confirmed by crystal structure
- c. Next, protein was prepared by Prepare protein protocol in Discovery Studio (DS) 2020 and then the binding site of 6K0J was defined form PDB site econds for further docking

A1. Thank you very much for your great suggestions! We revised these sentences to make it easier to be read and understood. Moreover, we revised the whole manuscript. The mentioned examples were revised as below.

- a. Moreover, the reported small-molecule DYRK2 inhibitors lack selectivity and exhibit poor druggability.
- b. and the detailed interactions between YK-2-69 and DYRK2 were further demonstrated by their co-crystal structure.
- c. The co-crystal structure of DYRK2 (PDB ID: 6K0J) was downloaded from RCSB Protein Data Bank, and the disordered conformations, crystal water, and ligand in the co-crystal were removed. The protein was applied to the CHARMM force field and prepared via *Prepare Protein* protocol of DS 2020. The binding site was defined by *From PDB Site Records* protocol for further molecule docking.

Q2. Figure caption 1 is unclear and does not explain the figure. Abbreviations colour-codes in 1A, B,

and C, are not defined.

A2. Thank you very much for your professional suggestions! We provided a more detailed caption of Figure 1 to explain the figure. Abbreviations colour-codes in 1A, B, and C, were also defined.

Fig. 1 DYRK2 is highly expressed in PCa. **a-c** Comparison of DYRK2 expression between tumor (red, n = 51) and normal (gray, n = 51) tissues (**a**), high (red, n = 345) and intermediate (blue, n = 152) risk PCa patients (**b**), age ≤ 65 (blue, n = 354) or > 65 (yellow, n = 143) PCa patients (**c**) in TCGA database. FPKM: fragment per kilobase million. **d** Kaplan-Meier survival plot of high (red line, n = 246) and low (blue line, n = 246) DYRK2 expression PCa patients. Log-rank test, $P = 0.015$. **e, f**

Analysis of DYRK2 expression in three PCa patients. DYRK2 mRNA level (e) and immunohistochemical analysis of DYRK2 expression (f) in tumor and normal tissues. Unpaired two-tailed Student's t test. Error bar, mean \pm SD, n = 3. g DYRK2 protein levels in different PCa cell lines, and normal prostate epithelial cell RWPE-1 was used as the control.

*, $P < 0.05$; **, $P < 0.01$; ***, $P < 0.001$; ****, $P < 0.0001$.

Q3. Not being an expert in Immunohistochemistry, I don't understand what figure 1E is supposed to show?

A3. Thank you very much! Immunohistochemistry was used to detect the expression of DYRK2 in prostate cancer tissues and adjacent normal tissues. Based on the result of staining with DYRK2 antibody, the tumor tissues showed the higher DYRK2 protein intensity than adjacent normal tissues, which means the higher expression of DYRK2 in prostate cancer tissues.

Q4. Likewise, the cell lines in Figure 1F are not explained, I can only guess that PC-3 and DU145 are the PC cell lines, clearly, only DU145 shows significantly enhanced mRNA levels. Why do some of the columns have NA on top, others some stars based on a statistical analysis that is not explained. Finally, 1G shows only one control cell line, also the figure is very unclear on the last hand top side - supposedly, the key area. I would expect a repeat with more cell lines and the display of the full western blot and not just a very selected area.

A4. Thank you very much for your great suggestions! The analysis of these 11 different cell lines was the preliminary exploration of the DYRK2 expression in different cancer lines. NA should be ns, which means "not significant". Our study was focused on prostate cancer, therefore, mRNA levels of DYRK2 in different cell lines (Figure 1F) was replaced. Meanwhile, protein levels of DYRK2 in more prostate cancer cells DU145, PC-3, 22Rv1, and LNCaP were tested. The results were shown in **Fig. 1g**.

We are very grateful for your kind reminder on the quality about western blotting. The quality of western blotting has been greatly improved in the new revision.

Fig. 1 DYRK2 is highly expressed in PCa. (g) DYRK2 protein levels in different PCa cell lines. Normal prostate epithelial cell RWPE-1 was used as the control.

Q5. The Methods part should describe the cell-culture of all cell lines mentioned here in the results section. Again, there are virtually no references in the methods section.

A5. Thank you very much for your great suggestions! The cell-culture methods of cell lines mentioned in the results section were all described in the Methods part. Moreover, references were cited in the methods section.

“**Cell culture.** The PCa DU145, 22Rv1, and LNCaP cell lines were obtained from American Type Culture Collection (ATCC), and cultured in endotoxin-free RPMI1640 supplemented with 10% fetal bovine serum (FBS, Gibco)⁶⁸. The DU145 shNC/shDYRK2 and 22Rv1 shNC/shDYRK2 cells were also cultured in endotoxin-free RPMI1640 with 10% FBS. PC-3 human PCa cells (ATCC), were cultured in F-12K with 10%FBS. RWPE-1 normal epithelial prostatic cells (ATCC) were cultured in KM supplemented with 10% FBS⁶⁹. All the cells are not among commonly misidentified cell lines, and were tested for mycoplasma contamination annually using a PCR Mycoplasma Detection Kit (G238, Applied Biological Materials Inc.). To prevent potential contamination, all the media were supplemented with Penicillin-Streptomycin (C0222, Beyotime) and Plasmocin prophylactic (ant-mpp, InvivoGen) according to the manufacturer’s instructions.”

Reference

68. Pan, X.W., et al. SMC1A promotes growth and migration of prostate cancer in vitro and in vivo. *Int. J. Oncol.* **49**, 1963-1972 (2016).

69. De Petrocellis, L., et al. Non-THC cannabinoids inhibit prostate carcinoma growth in vitro and in vivo: pro-apoptotic effects and underlying mechanisms. *Brit. J. Pharmacol* **168**, 79-102 (2013).

Q6. Figure 2 caption, again leaves a lot of information out. The Western blots again show a very

small selected area and are hence difficult to judge.

A6. Thank you very much for your professional suggestions! The more detailed information was added in the Fig. 2 caption.

We are very grateful for your kind reminder on the quality about western blotting. The quality of western blotting has been greatly improved in this revision.

Fig. 2 Knock-down of DYRK2 inhibited PCa *in vitro* and *in vivo*. **a, b** Protein level of DYRK2 in DU145 shNC/shDYRK2 (**a**) and 22Rv1 shNC/shDYRK2 (**b**) cells. **c, d** Cell viability of DU145 shNC/shDYRK2 cells (**c**) and 22Rv1 shNC/shDYRK2 cells (**d**) during a 5-day course. **e, f** Cell cycle

phase distribution of DU145 shNC/shDYRK2 cells (e) and 22Rv1 shNC/shDYRK2 cells (f) determined by flow cytometry g, h Apoptosis of DU145 shNC/shDYRK2 cells (g) and 22Rv1 shNC/shDYRK2 cells (h) determined by flow cytometry. i Western blotting analysis of indicated proteins in DU145 shNC and shDYRK2 cells. j, k BALB/c mice were implanted subcutaneously with DU145 shNC (n = 6) and shDYRK2 (n = 10) cells. Tumor volume of mice (j) was measured every two days. The shNC group was euthanized at 29th day and shDYRK2 group was euthanized at 49th day. Tumor tissues of mice treated with DU145 shDYRK2 and shNC cells were taken out, then the total proteins in the tumor were extracted and subjected to the western blotting analysis of indicated proteins (k).

*, $P < 0.05$; **, $P < 0.01$; ***, $P < 0.001$; ****, $P < 0.0001$; Unpaired two-tailed Student's t test. Error bar, mean \pm SD, n = 3.

Q7. It is not clear to the non-expert reader what figure 2 C, D, E, (left hand side) actually show.

A7. Thank you very much for your professional suggestions! We have provided more description in the caption to make it easy for reader to understand. The figures 2 C, D, E were renumbered as Supplementary Fig. 2 b, c, d, which were shown as below.

Supplementary Fig.2 DYRK2 KD significantly inhibited PCa. a DYRK2 mRNA level of in DU145 shNC/shDYRK2 cells and 22Rv1 shNC/shDYRK2 cells **b** Quantification of DU145

shNC/shDYRK2 cells and 22Rv1 shNC/shDYRK2 cells colony numbers after growth for ten days. **c**, **d** Quantification of migration (**c**) and invasion (**d**) ability of DU145 shDYRK2/shNC cells and 22Rv1 shDYRK2/shNC cells.

*, $P < 0.05$; **, $P < 0.01$; ***, $P < 0.001$; ****, $P < 0.0001$; unpaired Student's t test. Error bars, mean \pm SD, $n = 3$

A8. The virtual screening procedure should have been explained in a bit more detail in the results section, for example the exclusion criteria such as Lipinski's rule should have been explained here (as this is quite a crude measurement). Also, the choice of the starting library and the property of this library should be explained. Are these random chemicals or enriched as kinase inhibitors? Which target functions were used and what were the scores of the top compounds? And which experiment was used to derive an IC_{50} value of 263 nM?

A8. Thank you very much for your professional suggestions! We provided more information about virtual screening procedure in the structure-based virtual screening method section, including the explanation of Lipinski's rule, the choice and property of the starting library, the used target functions and the scores of the top compounds. The detailed virtual screening procedures were in Supplementary Experimental Procedures. The detailed information about top compounds was summarized in **Supplementary Table 1**.

DYRK2 kinase activity assays were carried out at Reaction Biology Corporation (<https://www.reactionbiology.com/>). The detailed kinase activity evaluation method was provided in DYRKs kinase activity and kinase-inhibitor specificity profiling assays method section.

“Structure-based virtual screening. Specs database (<http://www.specs.net>) contains more than 200,000 single synthesized, well-characterized and drug-like small molecules, and they are easy to obtain. It is a great choice for us to find a lead compound. Meanwhile, an in-house compound library has been established, which is mainly made up of kinase inhibitors. Therefore, the Specs database (221,097 compounds) and in-house compound library (about 3,000 compounds) were combined for the virtual screening. To eliminate compounds with poor drug-like properties, the combined ligand database was employed into DS 2020 and then filtered based on ‘Lipinski Rule of Five’¹, ‘Veber Rule’², and ‘Pan Assay Interference Compounds (PAINS)’^{3,4}. ‘Lipinski Rule of Five’ is used to

reserve drug-like ligands which have the following features: no more than 5 hydrogen bond donors, no more than 10 hydrogen bond acceptors, molecular weight no more than 500, and LogP no more than 5. The parameter “Number of Violations Allowed” in the ‘Lipinski Rule of Five’ is 1, which means that the ligands with at least three features mentioned can be reserved. ‘Veber Rule’ is used to select drug candidates with good oral bioavailability, which have the following features: no more than 10 rotatable bonds, polar surface area no more than 1,400 Å², or no more than 12 hydrogen bond donors and acceptors. ‘PAINS’ is used to remove ligands that are not drug-like or lead-like. After the elimination of ligands with poor drug-like properties, 195,483 compounds were reserved, which were then prepared through the *Prepare Ligands* protocol of DS 2020 to add hydrogens, remove duplicates, apply the CHARMM force field and minimize. The prepared ligands can be used for the following virtual screening.

The co-crystal structure of DYRK2 (PDB ID: 6K0J) was downloaded from RCSB Protein Data Bank⁵, and the disordered conformations, crystal water, and ligand in the co-crystal were removed. The protein was applied to the CHARMM force field and prepared via *Prepare Protein* protocol of DS 2020. The binding site was defined by *From PDB Site Records* protocol for further molecule docking.

The structure-based virtual screening procedure contains two programs, LibDock and CDOCKER. LibDock is a molecular docking program based on a high-throughput algorithm, which matches the poses of the ligand according to the structural feature of the receptor⁶. CDOCKER program employs high-temperature molecular dynamics to generate conformation and random rotation of the ligand⁷. Compared with Libdock, CDOCKER is more precise, but it needs more time. Therefore, Libdock was firstly used for primary screening, then CDOCKER was used for further screening.

195,483 prepared compounds firstly docked with DYRK2 protein via the Libdock protocol of DS2020. The pose with the highest score of each ligand was picked out, and ligands ranked based on the ‘Libdock Score’. Then 9,696 ligands with a LibDock score of more than 125 were remained, which were further filtered through CDOCKER. Conformation of each ligand was reserved in lowest energy and ligands ranked according to ‘-CDOCKER INTERACTION ENERGY’. Finally, 2,724 ligands with a “- CDOCKER INTERACTION ENERGY” of more than 55 remained.

Find Diverse Molecules protocol is used to find a diverse subset of ligands from the input ligands. Therefore, to further select ligands from 2,724 ligands, these ligands were clustered into 100 clusters

via *Find Diverse Molecules* protocol based on the functional-class fingerprint of diameter 6 (FCFP_6) properties. From 100 clusters, 15 compounds (Supplementary Fig. 3) were selected through visual inspection. In the end, the DYRK2 inhibitory activity of these 15 compounds was evaluated. Among these 15 compounds, compound **12** exhibited the best inhibitory activity against DYRK2 with an IC₅₀ value of 263 nM, and IC₅₀ values of the other 14 compounds were all more than 10 μM.”

“**Supplementary Table 1: Virtual screening and biological assay of 15 candidate hits^a.**

No	Score 1	Score 2	MW	LogP	H Donors/ Acceptors	IC ₅₀ (μM)
1	127.32	56.92	478.64	7.59	0/2	> 10
2	140.06	56.22	494.62	7.72	1/3	> 10
3	131.23	61.67	480.99	7.45	1/5	> 10
4	141.63	55.19	580.73	4.26	0/10	> 10
5	132.21	57.18	478.54	5.21	1/8	> 10
6	135.98	56.50	520.62	4.52	0/8	> 10
7	146.32	57.95	561.68	4.60	1/9	> 10
8	131.01	57.47	488.29	2.93	0/10	> 10
9	126.54	55.25	487.61	5.99	3/6	> 10
10	136.95	61.37	499.31	3.04	1/10	> 10
11	130.86	55.06	498.92	4.63	1/8	> 10
12	140.30	63.44	463.58	4.50	1/7	0.263
13	128.06	58.36	474.60	3.86	2/8	> 10
14	125.14	55.49	465.59	3.16	1/8	> 10
15	125.94	55.69	458.49	3.42	1/10	> 10
Staurosporine						0.179

^aScore 1: Libdock Score; Score 2: -CDOCKER INTERACTION ENERGY; MW: Molecular Weight;

H Donors/ Acceptors: Number of hydrogen bond donors and acceptors in Lipinski rule;

Staurosporine is the positive control in DYRK2 kinase assays.”

“**DYRKs kinase activity and kinase-inhibitor specificity profiling assays.** DYRKs kinase activity assays were carried out at Reaction Biology Corporation (<https://www.reactionbiology.com/>). For evaluating the inhibitory effect of tested compounds on DYRKs activity, we first prepared reaction buffer containing 20 mM HEPES (pH 7.5), 10 mM MgCl₂, 1 mM EGTA, 0.01% Brij35, 0.02 mg/mL BSA, 0.1 mM Na₃VO₄, 2 mM DTT, and 1% DMSO. Subsequently, DYRKs kinase and DYRKtide substrate were added into the reaction buffer and mixed gently. After test compounds were delivered

into the reaction mixture, ^{33}P -ATP (specific activity 0.01 $\mu\text{Ci}/\mu\text{L}$ final) was added to initiate the reaction. Then, the reactions were spotted onto P81 ion exchange paper (Whatman # 3698-915) followed by incubation for 2 h at room temperature. The filters were washed extensively in 0.75% phosphoric acid. Finally, measure the radioactive phosphorylated substrate remaining on the filter paper and calculate the remaining DYRKs activities in the tested compound group relative to the DMSO group.”

Q9. Is the description of interactions from line 160 on based on the docked (optimized?) structure or the co-crystal structure? In addition, the methods section lacks all key references. When referring to PDB files the underlying peer-reviewed publication should be cited.

A9. Thank you very much! The description of interactions between compound **12** and DYRK2 is based on the optimized docking structure, and the description of interactions between YK-2-69 and DYRK2 is based on the co-crystal structure. To avoid ambiguity, we have marked it (docked structure or the co-crystal structure) in the manuscript.

We are very grateful to you for your reminder on the correct citation of references, and we have cited all key references in the methods section, especially when referring to PDB files.

“**Structure-based virtual screening.** Specs database (<http://www.specs.net>) contains more than 200,000 single synthesized, well-characterized and drug-like small molecules, and they are easy to obtain. It is a great choice for us to find a lead compound. Meanwhile, an in-house compound library has been established, which is mainly made up of kinase inhibitors. Therefore, the Specs database (221,097 compounds) and in-house compound library (about 3,000 compounds) were combined for the virtual screening. To eliminate compounds with poor drug-like properties, the combined ligand database was employed into DS 2020 and then filtered based on ‘Lipinski Rule of Five’¹, ‘Veber Rule’², and ‘Pan Assay Interference Compounds (PAINS)’^{3,4}. ‘Lipinski Rule of Five’ is used to reserve drug-like ligands which have the following features: no more than 5 hydrogen bond donors, no more than 10 hydrogen bond acceptors, molecular weight no more than 500, and LogP no more than 5. The parameter “Number of Violations Allowed” in the ‘Lipinski Rule of Five’ is 1, which means that the ligands with at least three features mentioned can be reserved. ‘Veber Rule’ is used to select drug candidates with good oral bioavailability, which have the following features: no more than 10 rotatable bonds, polar surface area no more than 1,400 \AA^2 , or no more than 12 hydrogen

bond donors and acceptors. 'PAINS' is used to remove ligands that are not drug-like or lead-like. After the elimination of ligands with poor drug-like properties, 195,483 compounds were reserved, which were then prepared through the *Prepare Ligands* protocol of DS 2020 to add hydrogens, remove duplicates, apply the CHARMM force field and minimize. The prepared ligands can be used for the following virtual screening.

The co-crystal structure of DYRK2 (PDB ID: 6K0J) was downloaded from RCSB Protein Data Bank⁵, and the disordered conformations, crystal water, and ligand in the co-crystal were removed. The protein was applied to the CHARMM force field and prepared via *Prepare Protein* protocol of DS 2020. The binding site was defined by *From PDB Site Records* protocol for further molecule docking.

The structure-based virtual screening procedure contains two programs, LibDock and CDOCKER. LibDock is a molecular docking program based on a high-throughput algorithm, which matches the poses of the ligand according to the structural feature of the receptor⁶. CDOCKER program employs high-temperature molecular dynamics to generate conformation and random rotation of the ligand⁷. Compared with Libdock, CDOCKER is more precise, but it needs more time. Therefore, Libdock was firstly used for primary screening, then CDOCKER was used for further screening.

195,483 prepared compounds firstly docked with DYRK2 protein via the Libdock protocol of DS2020. The pose with the highest score of each ligand was picked out, and ligands ranked based on the 'Libdock Score'. Then 9,696 ligands with a LibDock score of more than 125 were remained, which were further filtered through CDOCKER. Conformation of each ligand was reserved in lowest energy and ligands ranked according to '-CDOCKER INTERACTION ENERGY'. Finally, 2,724 ligands with a "- CDOCKER INTERACTION ENERGY" of more than 55 remained.

Find Diverse Molecules protocol is used to find a diverse subset of ligands from the input ligands. Therefore, to further select ligands from 2,724 ligands, these ligands were clustered into 100 clusters via *Find Diverse Molecules* protocol based on the functional-class fingerprint of diameter 6 (FCFP_6) properties. From 100 clusters, 15 compounds (Supplementary Fig. 3) were selected through visual inspection. In the end, the DYRK2 inhibitory activity of these 15 compounds was evaluated. Among these 15 compounds, compound **12** exhibited the best inhibitory activity against DYRK2 with an IC₅₀ value of 263 nM, and IC₅₀ values of the other 14 compounds were all more than 10 μM."

References

1. Lipinski, C.A., Lombardo, F., Dominy, B.W. & Feeney, P.J. Experimental and computational approaches to estimate solubility and permeability in drug discovery and development settings. *Adv. Drug. Deliver. Rev.* **23**, 3-25 (1997).
2. Veber, D.F., et al. Molecular properties that influence the oral bioavailability of drug candidates. *J. Med. Chem.* **45**, 2615-2623 (2002).
3. Walters, W.P. & Murcko, M.A. Prediction of 'drug-likeness'. *Adv. Drug. Deliver. Rev.* **54**, 255-271 (2002).
4. Hann, M., et al. Strategic pooling of compounds for high-throughput screening. *J. Chem. Inf. Comp. Sci.* **39**, 897-902 (1999).
5. Banerjee, S., et al. Ancient drug curcumin impedes 26S proteasome activity by direct inhibition of dual-specificity tyrosine-regulated kinase 2. *Proc. Natl. Acad. Sci. U S A* **115**, 8155-8160 (2018).
6. Diller, D.J. & Merz, K.M. High throughput docking for library design and library prioritization. *Proteins* **43**, 113-124 (2001).
7. Wu, G.S., Robertson, D.H., Brooks, C.L. & Vieth, M. Detailed analysis of grid-based molecular docking: A case study of CDOCKER - A CHARMM-based MD docking algorithm. *J. Comput. Chem.* **24**, 1549-1562 (2003).

Q10. The authors should also explain what the “systematic structure optimization” entailed. This step is usually the most challenging one.

A10. Thank you very much for your professional suggestions! We provided all the details about the optimization process of compound **12** to obtain compound YK-2-69 in the Methods section.

“**Structural optimization.** To obtain more potent DYRK2 inhibitor, structural optimization on the lead compound **12** was conducted (Supplementary Fig. 4). Firstly, the isopropyl group on the piperazine was replaced with different sizes substituents (**16-18**), among which compounds **16** and **18** with a small ethyl or hydrogen group showed similar activity as lead **12**, while compound **17** with a big Boc group exhibited significantly decreased activity. When benzothiazole was changed to pyrazolo[1,5-a]pyrimidine (**19**), DYRK2 inhibitory activity significantly decreased. Therefore, benzothiazole was retained in the following structural optimization. To our delight, the optimization on the linker between pyridine and piperazine (**20**: 35 nM and **21**: 85 nM) remarkably improved DYRK2 inhibitory activity, especially the carbonyl linker. Subsequently, the change of pyridine to

pyrimidine (**22**: 697 nM) also decreased DYRK2 inhibitory activity. Therefore, pyridine is a favorable moiety for the maintenance of activity. After confirming the benzothiazole, pyridine, and carbonyl linker as the optimized groups, the subsequent modification was carried out by introducing substituents on the benzothiazole. The introduction of methyl substituent generated **23** and **24**, which exhibited stronger inhibition on DYRK2 with IC₅₀ values of 22 and 27 nM, respectively. The introduction of dimethylamine substituent (**25**, **26**) further improved DYRK2 inhibitory activity, and **26** exhibited the most potent inhibition on DYRK2 with an IC₅₀ value of 9 nM. The change of pyrimidine to thieno[3,2-d]pyrimidine generated **27**, which showed no inhibition on DYRK2. Taken together, **26** was confirmed as the most potent DYRK2 inhibitor, which was re-named as YK-2-69 and selected for further biological evaluation.”

Supplementary Fig. 4 Chemical structures and DYRK2 kinase IC₅₀ values of compounds 16-27.

To further improve DYRK2 inhibitory activity, representative derivatives **16-27** were synthesized and further tested their inhibitory activity against DYRK2 kinase. Among them, compound **26** (re-named as YK-2-69) exhibited the most potent DYRK2 inhibitory activity with an IC₅₀ value of 9 nM.”

Figure 3

Q11. The steps from compounds **12** to YK-2-69 should be explained.

A11. Thank you very much for your professional suggestions! We provided all the details about the optimization process of compound **12** to obtain compound YK-2-69 in the Methods section.

“**Structural optimization.** To obtain more potent DYRK2 inhibitor, structural optimization on the lead compound **12** was conducted (Supplementary Fig. 4). Firstly, the isopropyl group on the piperazine was replaced with different sizes substituents (**16-18**), among which compounds **16** and **18** with a small ethyl or hydrogen group showed similar activity as lead **12**, while compound **17** with a big Boc group exhibited significantly decreased activity. When benzothiazole was changed to pyrazolo[1,5-a]pyrimidine (**19**), DYRK2 inhibitory activity significantly decreased. Therefore, benzothiazole was retained in the following structural optimization. To our delight, the optimization on the linker between pyridine and piperazine (**20**: 35 nM and **21**: 85 nM) remarkably improved DYRK2 inhibitory activity, especially the carbonyl linker. Subsequently, the change of pyridine to pyrimidine (**22**: 697 nM) also decreased DYRK2 inhibitory activity. Therefore, pyridine is a favorable moiety for the maintenance of activity. After confirming the benzothiazole, pyridine, and carbonyl linker as the optimized groups, the subsequent modification was carried out by introducing substituents on the benzothiazole. The introduction of methyl substituent generated **23** and **24**, which exhibited stronger inhibition on DYRK2 with IC₅₀ values of 22 and 27 nM, respectively. The introduction of dimethylamine substituent (**25**, **26**) further improved DYRK2 inhibitory activity, and **26** exhibited the most potent inhibition on DYRK2 with an IC₅₀ value of 9 nM. The change of pyrimidine to thieno[3,2-d]pyrimidine generated **27**, which showed no inhibition on DYRK2. Taken together, **26** was confirmed as the most potent DYRK2 inhibitor, which was re-named as YK-2-69 and selected for further biological evaluation.”

Supplementary Fig. 4 Chemical structure and DYRK2 kinase IC₅₀ values of compounds 16-27.

To further improve DYRK2 inhibitory activity, representative derivatives **16-27** were synthesized and further tested their inhibitory activity against DYRK2 kinase. Among them, compound **26** (re-named as YK-2-69) exhibited the most potent DYRK2 inhibitory activity with an IC₅₀ value of 9 nM.”

Q12. The gels should be shown as complete.

A12. Thank you very much for your professional suggestions! We are very grateful for your kind reminder on the quality about western blotting. The quality of western blot has been greatly improved in this revision.

Fig. 3 Discovery of the highly selective DYRK2 inhibitor YK-2-69. (e) Validation of interaction of P-4E-BP1 (Thr37/46) with DYRK2 by co-immunoprecipitation assay. 1% volume of cell lysate was

used as input. **f** Western blotting analysis of indicated proteins after treatment with DMSO or YK-2-69 for 48 h. The special bands of 4E-BP1 and p-4E-BP1 were shown with arrows.

Q13. 3c is pleasing to the eye but it is unclear what actually is shown. What are the scales and what are the color codes?

A13. Thank you very much for your professional suggestions! **Figure 3c** is renamed as **Figure 3d**. We followed your great advice to add the legend in **Figure 3d**. The size of dot represents the inhibition rate. We marked DYRK2 as red, and other targets as blue.

“**Figure 3. Discovery of the Highly Selective DYRK2 Inhibitor YK-2-69.** (d) Kinase selectivity experiment of YK-2-69 (1 μ M) was carried out at Reaction Biology Corporation (<https://www.reactionbiology.com/>). The dot size indicates the inhibitory rate. A dot represents a type of kinase and the red dot represents DYRK2.”

Q14. Figure D should be a ribbon diagram and E should show unbiased electron density of the ligand. Once again, the method section lacks all relevant references, including the one for the detector (Pilatus I assume), flash-freezing, structure solution, refinement. As the protein seems to have two independent molecules, I also assume local NCS restraints were used. Molprobvity and Phenix references (peer-reviewed publications) must be included. In the crystallographic table B-factors for

protein, ligand and waters should be given.

A14. Thank you very much! We followed your professional suggestions to better display the co-crystal of YK-2-69 with DYRK2.

We are very grateful for your professional advice to cite the relevant references. We have added all relevant references, including the one for the detector (Pilatus I assume), flash-freezing, structure solution, refinement, Molprobity and Phenix in the method section of our manuscript.

Moreover, B-factors for protein, ligand and waters have been added in the crystallographic table (**Supplementary Table 2**).

“Co-crystallization, data collection, and structure determination. DYRK2⁷²⁻⁴⁷⁹ (10 mg/mL) was incubated with 1 mM YK-2-69 at 4 °C before crystallization. The protein-YK-2-69 mixture was then mixed in a 1:1 ratio with crystallization solution (0.1 M sodium citrate pH 5.5, 8% PEG3350) in a final drop size of 2 µL. The initial DYRK2-YK-2-69 crystals were grown at 4 °C by the hanging-drop vapor diffusion method and optimized by seeding. Cuboid-shaped crystals appeared after 2-7 days. Crystals were cryoprotected in the crystallization solution supplemented with 30% glycerol before being frozen in liquid nitrogen. The X-ray diffraction data were collected using a PILTUS3 6M detector on beamline BL19U at Shanghai Synchrotron Radiation Facility (SSRF)⁶¹. The diffraction data were indexed, integrated, and scaled using HKL-2000 (HKL Research). The structure was determined by molecular replacement using the published DYRK2 structure (PDB ID: 5LXD) as the search model using the Phaser-MR program in Phenix^{47,62}. A clear electron density was observed in the center of the ATP binding pocket after molecular replacement. YK-2-69 was fitted using the eBLOW and LigandFit program in Phenix^{63,64}. The structural model was further adjusted in Coot and refined using Phenix while using NCS restraints^{65,66}. The quality of the structural model was checked using the MolProbity program in Phenix⁶⁷. The crystallographic data and refinement statistics are summarized in Supplementary Table 2.”

References

61. Zhang, W.-Z., et al. The protein complex crystallography beamline (BL19U1) at the Shanghai Synchrotron Radiation Facility. *Nucl. Sci. Tech.* **30**, 170 (2019).
62. McCoy, A.J., et al. Phaser crystallographic software. *J. Appl. Crystallogr.* **40**, 658-674 (2007).
63. Moriarty, N.W., Grosse-Kunstleve, R.W. & Adams, P.D. electronic Ligand Builder and

Optimization Workbench (eLBOW): a tool for ligand coordinate and restraint generation. *Acta Crystallogr. D Biol. Crystallogr.* **65**, 1074-1080 (2009).

64. Adams, P.D., et al. PHENIX: a comprehensive Python-based system for macromolecular structure solution. *Acta Crystallogr. D Biol. Crystallogr.* **66**, 213-221 (2010).

65. Emsley, P., Lohkamp, B., Scott, W.G. & Cowtan, K. Features and development of Coot. *Acta Crystallogr. D Biol. Crystallogr.* **66**, 486-501 (2010).

66. Afonine, P.V., et al. Towards automated crystallographic structure refinement with phenix.refine. *Acta Crystallogr. D Biol. Crystallogr.* **68**, 352-367 (2012).

67. Williams, C.J., et al. MolProbity: More and better reference data for improved all-atom structure validation. *Protein Sci.* **27**, 293-315 (2018).

“**Fig. 4 Co-crystal structure of YK-2-69 with DYRK2.** **a** The F_0-F_C omitted map is contoured at 3.0σ and shown as a blue mesh, which reveals the presence of YK-2-69. DYRK2 is shown as ribbons. PDB ID: 7EJV. **b** YK-2-69 occupies the ATP binding pocket of DYRK2. DYRK2 is shown as surface, and YK-2-69 is shown as spheres. **c** Detailed interactions between YK-2-69 and DYRK2 in co-crystal structure.”

“Supplementary Table 2: Co-crystal data collection, phasing and refinement statistics

DYRK2-YK-2-69	
Data Collection	
Beam Line	BL19U, SSRF
Space group	C 2 2 21
Cell dimensions	
a , b , c (Å)	60.86 129.387 291.733
α , β , γ (°)	90 90 90
wavelength	0.9785
Resolution Limits(Å)	19.86 – 2.5 (2.589 – 2.5)
No. Unique reflections	40358 (3967)
Completeness (%)	99.70 (99.97)
CC1/2	0.998 (0.696)
R_{merge}	0.1797 (1.907)
$I/\sigma I$	14.02 (1.44)
Completeness (%)	99.70 (99.97)
Refinement	
No. reflections	40351
$R_{\text{work}} / R_{\text{free}}$	0.2115 / 0.2517
R.m.s.d for bonds (Å)	0.010
R.m.s.d for angles(°)	1.31
Number of non-hydrogen atoms	6351
Macromolecules	6279
Ligands	72
Solvent	0
B-factor	
Averaged	70.35
Macromolecules	70.44
Ligands	62.4
Ramachandran plot (%)	
Favored (%)	96.47
Allowed (%)	3.53
Outliers	0.00

Statistics for the highest-resolution shell are shown in parentheses.”

Q15. The crystal structure determination is of appropriate quality, however, it is difficult to judge the quality of the ligand position as no unbiased electron density is given. I would have liked to have seen a biophysical method (TSA, ITC or SPR) to give at least an indication of binding or better a proper binding constant.

A15. Thank you very much! We followed your great suggestion to determine the K_d values of compound **12** and YK-2-69 through the MST method. The K_d values of compound **12** and YK-2-69 with DYRK2 were 4.21 μM and 95 nM, respectively. The results were shown in the **Supplementary Fig. 6**.

“**Supplementary Fig. 6** The binding constants of compound **12** and YK-2-69 with DYRK2. **a, b** The binding constants of compound **12** (**a**) and YK-2-69 (**b**) with DYRK2 were determined by Microscale Thermophoresis.”

Q16. There are 11 co-crystal structures in the database, so the authors should compare their binding mode with what had been observed before.

A16. Thank you very much for your professional suggestion! We analyzed the binding modes of these compounds and YK-2-69 with DYRK2. They all bind to the ATP-binding pocket of DYRK2.

Lys-231 plays a significant role in the interaction between compounds and DYRK2. The pyrimidine ring and secondary amine of YK-2-69 interacted with Lys-231 in the form of two hydrogen bonds. In other co-crystal structures (PDB ID: 3KVW, 4AZF, 6HDR), Lys-231 also plays an important role in the formation of hydrogen bonds.

Q17. When reporting IC₅₀ values the authors ought to consider significant numbers, and give realistic values.

For example: "... against the proliferation of PC-3 and DU145 cell lines with IC₅₀ values of 1.069 and 3.302 μM, respectively (Fig. 4a)."

A17. Thank you very much for your great advice! We gave realistic values when reporting IC₅₀ values in the manuscript.

Supplementary Fig. 7 YK-2-69 suppressed cell proliferation and metastasis and promoted apoptosis in PCa cells. a-c Antiproliferative activities of YK-2-69 (a), palbociclib (b) and enzalutamide (c) against of DU145, PC-3, 22Rv1, and RWPE-1 cell lines.

Q18. Figure 4 and 5 again contain a lot of information that is not sufficiently explained in the figure caption. I personally find the tendency to cut out single gel bands from gel somewhat disconcerting.

A18. Thank you very much! We followed your great advice to provide more information in the figure captions to make them sufficiently explained. Moreover, we are very grateful for your kind reminder on the quality about western blotting. The quality of western blotting has been greatly improved in this revision. **Fig. 4** and **5** were renamed **Fig. 5** and **6** in this revision, which were shown as below. The improved western blotting was in **Supplementary Fig. 7**, which was also shown as below.

Fig. 5 YK-2-69 inhibited cell growth and metastasis and induced apoptosis in Pca cells.

a Antiproliferative activity of YK-2-69 against DU145 and 22Rv1 shDYRK2 cells. **b, c** Effects of YK-2-69 (4, 8, 12 μM or 0.5, 1, 2 μM), palbociclib (2 or 0.5 μM), or enzalutamide (40 or 20 μM) treatment on the DU145 (**b**) and 22Rv1 (**c**) cells viability during a 5-day course. **d, e** Quantification of DU145 (**d**) and 22Rv1 (**e**) cells colony numbers. Before being plated on the 24-well plate for colony formation, DU145 and 22Rv1 cells were treated with DMSO, YK-2-69 (2, 4, 8 or 0.5, 1, 2

μM), palbociclib (2 or 0.5 μM), or enzalutamide (40 or 20 μM) for 48 h. **f, g** Quantification of migration ability of DU145 (**f**) and 22Rv1(**g**) cells after treatment with DMSO, YK-2-69 (2, 4, 8 μM or 0.5, 1, 2 μM), palbociclib (2 or 0.5 μM), or enzalutamide (40 or 20 μM) for 48 h. **h, i** Quantification of invasion ability of DU145 (**h**) and 22Rv1(**i**) cells after treatment with DMSO, YK-2-69 (2, 4, 8 or 0.5, 1, 2 μM), palbociclib (2 or 0.5 μM), or enzalutamide (40 or 20 μM) for 48 h. **j, k** Apoptosis of DU145 (**j**) and 22Rv1 (**k**) cells after treatment with DMSO, YK-2-69 (2, 4, 8 or 0.5, 1, 2 μM), palbociclib (2 or 0.5 μM), or enzalutamide (40 or 20 μM) for 48 h determined by flow cytometry. **l, m** Cell cycle phase distribution of DU145 (**l**) and 22Rv1 (**m**) cells after treatment with DMSO, YK-2-69 (2, 4, 8 or 0.5, 1, 2 μM), palbociclib (2 or 0.5 μM), or enzalutamide (40 or 20 μM) for 48 h determined by flow cytometry.

*, $P < 0.05$; **, $P < 0.01$; ***, $P < 0.001$; ****, $P < 0.0001$; Unpaired two-tailed Student's t test. Error bar, mean \pm SD, $n = 3$.

Supplementary Fig. 7 YK-2-69 suppressed cell proliferation and metastasis and promoted

apoptosis in PCa cells.

a-c Antiproliferative activities of YK-2-69 (**a**), palbociclib (**b**) and enzalutamide (**c**) against of DU145, PC-3, 22Rv1, and RWPE-1 cell lines. **d** Effects of YK-2-69 (1, 2, 4 μ M), palbociclib (4 μ M), or enzalutamide (40 μ M) treatment on the viability of the PC-3 cells during a 5-day course. **e** Quantification of PC-3 cells colony numbers. Before being plated on the 24-well plate for colony formation, PC-3 cells were treated with DMSO, YK-2-69 (1, 2, 4 μ M), palbociclib (2 μ M), or enzalutamide (40 μ M) for 48 h. **f, g** Quantification of migration (**f**) and invasion (**g**) ability of PC-3 after treatment with DMSO, YK-2-69(1, 2, 4 μ M), palbociclib (2 μ M), or enzalutamide (40) for 48 h. **h** Apoptosis of PC-3 cells after treatment with DMSO, YK-2-69 (1, 2, 4 μ M), palbociclib (2 μ M), or enzalutamide (40 μ M) for 48 h determined by flow cytometry. **i** Cell cycle phase distribution of PC-3 cells after treatment with DMSO, YK-2-69 (1, 2, 4 μ M), palbociclib (2 μ M), or enzalutamide (40 μ M) for 48 h determined by flow cytometry. **j** Western blotting analysis of indicated proteins in DU145 cells after treatment with DMSO or YK-2-69 (2, 4, 8 μ M) for 48 h.

*, $P < 0.05$; **, $P < 0.01$; ***, $P < 0.001$; ****, $P < 0.0001$; ns, not significant; Unpaired two-tailed Student's t test. Error bar, mean \pm SD, n = 3.

Fig. 6 Transcriptome-wide RNA sequencing assays in PCa DU145 cells. **a** Transcriptome strategy of RNA-sequencing conducted on DU145 cells exposed to YK-2-69 (3 μ M) for 48 h. The shNC, shDYRK2, DMSO, and YK-2-69 groups all contain two biological replicates. The principal component analysis was used to identify transcriptome differences between two samples. **b** Venn diagram of upregulated and downregulated signaling pathways in DYRK2 KD- and YK-2-69-treated DU145 cells. The number of genes in every signaling pathway is more than 50. Normalized enrichment score (NES) > 1 or < -1; $p < 0.05$; $FDR < 0.25$. **c** The signaling pathways enriched in different groups obtained through Gene set enrichment analysis (GSEA). **d** The core-enriched

decreased (blue) and increased (red) signaling pathways in shDYRK2 and YK-2-69 treatment groups when compared with shNC and DMSO groups, respectively. The signaling pathways with $p < 0.05$ are presented. **e** The relative abundance of genes involved in MYC target V1, MYC target V2 and Mitotic SPINDLE in DYRK2 KD- and YK-2-69-treated DU145 cells. **f** GSEA was used to analyze the effects of DYRK2 KD or YK-2-69 treatment on the ANDROGEN RESPONSE signaling pathway in DU145 cells. **g** Effects of DYRK2 KD or YK-2-69 treatment on the RRS1, GRWD1, CCNG2, and YPEL3 mRNA levels in DU145 cells.

Q19. I am not familiar with animal studies so I can't really comment on those, however, again I noticed that the authors should consider how many significant numbers they quote. It seems unlikely that any of the numbers have precision of 1/10000.

A19. Thank you very much for your great advice! We provided realistic values when reporting the results of animal studies in the manuscript, especially the results of pharmacokinetic parameters in Table 1.

Table 1 Pharmacokinetic parameters of compound YK-2-69 in SD rats^a

Compd.	Admin.	C _{max} (ng/mL)	AUC _{0-∞} (h*ng/mL)	MRT _{0-∞} (h)	T _{max} (min)	t _{1/2} (h)	CL (mL/h/kg)	F (%)
YK-2-69	IV	974	1,503	3.5	2	3	669	-
	PO	674	8,384	8.9	240	5	511	56

^aValues are the average of three runs. C_{max}, maximum concentration; AUC, area under the plasma concentration-time curve; MRT, mean residence time; t_{1/2}, half-life; CL, clearance; F, oral bioavailability. Dose: p.o. at 10 mg/kg. Dose: i.v. at 1 mg/kg

Reviewer 4:

Summary: In this manuscript, the authors identified DYRK2 as a therapeutic target gene in prostate cancer (PC). By analyzing the TCGA PC dataset, they showed upregulation of DYRK2 in PC compared with matched normal controls, and improved relapse-free survival in the DYRK2-low group. The knockdown of DYRK2 inhibited the tumor growth both in vitro and in vivo. Motivated by these observations, the authors developed a highly selective DYRK2 inhibitor (YK-2-69) with favorable safety properties. They demonstrated the effect of YK-2-69 was similar to that of DYRK2 knockdown based on the transcriptomic analysis on a PC cell line. Furthermore, YK-2-69 showed better anti-tumor activity than a first-line drug in the xenograft mouse model.

This manuscript is well written and nicely demonstrates how anti-cancer drug can be developed by combining multidisciplinary approaches.

Reply: Thank you very much for the positive comments! This reflects the merits of our manuscript! We completely followed your professional suggestions to conduct all the experiments and made a careful revision that helped us to improve our manuscript and to meet the scientific standard of this journal.

Major points:

Q1. Figure 1B-D: The authors showed that age is associated with the expression of DYRK2 in Figure 1C. I'm wondering whether the observed association in Figure 1B and 1D is still significant after accounting for age.

A1: Thank you very much! We followed your great advice to account for the age factor and further analyze the association of DYRK2 expression.

As shown in Figure lower left, the high risk group consists of 240 patients with the age ≤ 65 and 105 patients with the age > 65 ; the intermediate risk group consists of 114 patients with the age ≤ 65 and 38 patients with the age > 65 . Therefore, the ratio of patients with the age ≤ 65 to patients with the age > 65 is similar in the high and intermediate risk groups, which does not affect the analysis result.

For Figure 1d, we also compared the age factor in low and high DYRK2 groups. In Figure 1d, the low DYRK2 group consists of 184 patients with the age ≤ 65 and 62 patients with the age > 65 ; the high DYRK2 group consists of 167 patients with the age ≤ 65 and 79 patients with the age > 65 .

Moreover, as shown in Figure lower right, there was no significant difference of the age between low and high DYRK2 groups, which does not affect the analysis result.

Q2. Figure 5: The authors claimed that the effect of YK-2-69 was similar to that of DYRK2 knockdown by comparing differentially expressed genes and enriched signatures from RNA sequencing (RNA-seq) data. However, their main claim is not well supported by data presented. More systematic analysis is required.

Q2.1 First, as a negative control, samples treated with the first-line PC drug should be included.

A2.1: Thank you very much for your great advice! Following your professional suggestion, we added the first-line PC drug enzalutamide as the control and re-carried out the RNA-seq test. We found that DYRK2 KD and YK-2-69 treatment had no effects on ANDROGEN_RESPONSE signaling pathway, and enzalutamide caused the significant suppression of ANDROGEN_RESPONSE.

Q2.2 Second, for generalizability, one more PC cell line (PC-3?) highly expressing DYRK2 should be analyzed.

A2.2: Thank you very much for your great advice! We have analyzed RNA-Seq data of the other PC cell line 22Rv1, which highly expressed DYRK2.

Q2.3 Third, visualizing transcriptomic differences among samples using principal component analysis or similar methods might be helpful.

A2.3: Thank you very much for your professional suggestion! We used the R package to carry out principal component analysis. The principal component analysis demonstrated that the visualizing transcriptomic differences among samples were small. The result of principal component analysis was shown as below:

Q2.4 In addition, using upregulated or downregulated genes in shDYRK2 compared to shVehicle as a signature gene set, the authors can perform GSEA on a ranked gene list of YK-2-69 treated samples compared to WT.

A2.4: Thank you very much your great suggestion! We used downregulated genes in shDYRK2 compared to shVehicle as a signature gene set, then performed GSEA on a ranked gene list of YK-2-69 treated samples compared to WT. The results demonstrated that the genes which were downregulated due to DYRK2 KD were also downregulated when treated with YK-2-69.

The result of GSEA analysis was shown as below:

Q2.5 Finally, the authors should carefully analyze differentially expressed genes and signatures between shDYRK2 and YK-2-69 treated samples.

A2.5: Thank you very much for your great suggestion! We carefully re-analyzed differentially expressed genes between shDYRK2 and YK-2-69 treated samples, and found significant expression changes in human regulator ribosome synthesis 1 (RRS1), glutamate-rich WD40 repeat containing 1 (GRWD1), cyclin G2 (CCNG2), and Yippee-like-3 (YPEL3).

RRS1, an essential nuclear protein involved in ribosome biogenesis, is overexpressed in some human cancers, and downregulation of RRS1 causes a G1 arrest of cell cycle. GRWD1, a negative transcriptional regulator of P53, plays an oncogenic activity in human cancers. Meanwhile, previous studies have shown that overexpression of CCNG2 can induce apoptosis and inhibit cell proliferation. YPEL3, a P53-regulated gene, has been reported to display growth suppressive and EMT inhibitory activity. The experimental results of qRT-PCR demonstrated that oncogenes RRS1 and GRWD1 were downregulated and tumor suppressors CCNG2 and YPEL3 were upregulated regardless of knocking down DYRK2 or YK-2-69 treatment.

DU145

22Rv1

Minor points:

1. Line 86-87: “TCGA Cancer Genome Atlas” -> The Cancer Genome Atlas (TCGA)

A1. Thank you very much! We have revised “TCGA Cancer Genome Atlas” to “The Cancer Genome Atlas (TCGA)”

2. Line 156: “Considering the highly expression level” -> “Considering the high expression level”.

A2. Thank you very much! We have revised “Considering the highly expression level” to “Considering the high expression level”

3. Figure 3C: What do the color and size of dots mean?

A3. Thank you very much! A blue dot means a type of kinase and the dot size means inhibitory rate. We followed your great advice to add the detailed legend “The dot size indicates the inhibitory rate. A dot represents a type of kinase and the red dot represents DYRK2.”.

“**Figure 3. Discovery of the Highly Selective DYRK2 Inhibitor YK-2-69.** (d) Kinase selectivity experiment of YK-2-69 (1 μ M) was carried out at Reaction Biology Corporation (<https://www.reactionbiology.com/>). The dot size indicates the inhibitory rate. A dot represents a type of kinase and the red dot represents DYRK2.”

4. Figure 5A and Supplementary Figure 5A: The authors should evaluate the statistical significance of the overlaps.

A4. Really appreciate your professional suggestion! In the previous version, we just overlapped the genes ($|\text{Log}_2\text{FC}| > 1$ and $P < 0.05$) to generate Figure 5A. For Supplementary Figure 5A, we overlapped the genes ($\text{FDR} < 0.05$ and $|\text{FC}| > 1.5$) to generate it.

In the new version, we followed your great advice to analyze the upregulated and downregulated signaling pathways in DYRK2 KD- and YK-2-69-treated Du145 and 22Rv1 cells. The results were shown as Venn diagram. The number of genes in every signaling pathway is more than 50. Normalized enrichment score (NES) > 1 or < -1 ; $p < 0.05$; $\text{FDR} < 0.25$.

Figure 5A

Supplementary Figure 5A

Venn diagram of upregulated and downregulated signaling pathways

Du-145

22Rv1

Size > 50; |NES| > 1; p < 0.05; FDR < 0.25

5. Figure 5B: What does the x-axis mean?

A5: Thank you very much! The x-axis means “NES (Normalized Enrichment score)”, which has been added to the new version. The NES values of the pathways with $p < 0.05$ are presented.

6. Figure 5D and Supplementary Figure 5B: The authors should report P-values.

A6: Thank you very much! We have added P-values in the Figure 5D and Supplementary Figure 5B.

Figure 5D

Supplementary Figure 5B

7. Line 253-255 “Global genome-enrichment analysis showed that DYRK2 KD and YK-2-69 treatment exhibited similar effects on the regulation of a set of functionally important signaling pathways.”: The authors should provide data supporting this sentence.

A7: Thank you very much! We have followed your professional suggestion to revise this sentence and make it more accurate. “In summary, these data suggested that DYRK2 KD and YK-2-69 treatment played important and similar roles in cell proliferation inhibition.”

REVIEWER COMMENTS

Reviewer #1 CONFIDENTIAL COMMENTS: supports publication

Reviewer #2 (Remarks to the Author):

The authors have improved the manuscript following the reviewers' suggestions.

Reviewer #3 (Remarks to the Author):

Overall this is an excellent paper with important results. I feel that the authors have taken (almost) all suggestions into account and hence I am happy to recommend publication.

Reviewer #4 (Remarks to the Author):

The authors should specify or highlight changes in their manuscript and rebuttal. It is difficult for me to follow changes in this revised manuscript. The revised manuscript addressed most of my previous concerns except the following point:

1. From Fig. 6a, the authors concluded that transcriptomic differences among samples are small, but this is a misleading statement. Fig. 6a suggests that transcriptomic differences between shDYRK2 and YK-2-69 (captured by PC1 explaining 82.2% of variance) is much larger than differences between shDYRK2/YK-2-69 and controls (captured by PC2 explaining 10.9% of variance). This indicates that DYRK2 KD and YK-2-69 treatment might induce distinct transcriptomic changes and raises a concern about potential side effects of YK-2-69. This issue should be carefully examined by the authors. They should perform GO/pathway analysis on DEGs or GSEA between shDYRK2 and YK-2-69 and discuss the observed changes in a systematic way.

Minor points:

1. Line 312-313: "differentially expressed genes" -> "differentially expressed genes between shDYRK2 and YK-2-69 treated samples". The authors should also provide a volcano plot or supplementary table listing these DEGs.

** See Nature Portfolio's author and referees' website at www.nature.com/authors for information about policies, services and author benefits.

We would like to express our sincere thanks to the editor and reviewers for the positive and constructive comments!

Reviewer #4:

The authors should specify or highlight changes in their manuscript and rebuttal. It is difficult for me to follow changes in this revised manuscript. The revised manuscript addressed most of my previous concerns.

Reply: Thank you very much for the positive comments! We submitted two versions with highlight, which marked the changes of our manuscript for the first round and second round revision, respectively.

Moreover, we completely followed your professional suggestions to make a careful revision that helped us to improve our manuscript and meet the scientific standard of this journal.

Q1. From Fig. 6a, the authors concluded that transcriptomic differences among samples are small, but this is a misleading statement. Fig. 6a suggests that transcriptomic differences between shDYRK2 and YK-2-69 (captured by PC1 explaining 82.2% of variance) is much larger than differences between shDYRK2/YK-2-69 and controls (captured by PC2 explaining 10.9% of variance). This indicates that DYRK2 KD and YK-2-69 treatment might induce distinct transcriptomic changes and raises a concern about potential side effects of YK-2-69. This issue should be carefully examined by the authors. They should perform GO/pathway analysis on DEGs or GSEA between shDYRK2 and YK-2-69 and discuss the observed changes in a systematic way.

A1: Thank you very much for your professional suggestion! Firstly, we revised the misleading statement “transcriptomic differences among samples are small” in the manuscript.

As shown in Fig. 6a-e, many signaling pathways regulated by DYRK2 KD could also be regulated by YK-2-69 treatment. These data suggested that DYRK2 KD and YK-2-69 treatment played important and similar roles in cell proliferation inhibition.

Meanwhile, completely following your professional suggestion, we further performed GSEA to research the transcriptomic changes induced by DYRK2 KD and YK-2-69 treatment, which indicated that DYRK2 KD and YK-2-69 treatment induced some different transcriptomic changes. As shown in Fig. 6i, YK-2-69 treatment induced inhibition of DNA repair and TGF β signaling pathways, while DYRK2 KD had no remarkable effects on these two signaling pathways. Inhibition of DNA repair is a successful therapeutic strategy for cancers with several approved drugs in the market. TGF- β also plays a critical role as a tumor promoter in late-stage cancer, and a number of drugs for inhibiting TGF β signaling pathway have been developed and evaluated in clinical trials. Therefore, compared with DYRK2 KD, YK-2-69 may regulate some different signaling pathways to generate potent antitumor activity, while not lead to potential side effects. Furthermore, the excellent safety properties of YK-2-69 were also confirmed by the *in vivo* toxicity evaluation.

Fig. 6i The different signaling pathways enriched in DYRK2 KD and YK-2-69 treatment groups obtained through GSEA.

Minor points:

Q1. Line 312-313: “differentially expressed genes” -> “differentially expressed genes between shDYRK2 and YK-2-69 treated samples”. The authors should also provide a volcano plot or supplementary table listing these DEGs.

A1. Really appreciate your professional suggestion! We changed “differentially expressed genes” to “differentially expressed genes between shDYRK2 and YK-2-69 treated samples” in the manuscript.

Completely following your professional suggestion, we provided a volcano plot to display DEGs between shDYRK2 and YK-2-69 treated samples (Fig. 6f). Meanwhile, we submitted a supplementary file to show all these DEGs (Supplementary Dataset).

f

Fig. 6f Volcano plot of significantly affected genes (absolute fold change > 2, P value < 0.05) in DU-145 shDYRK2 group relative to shNC group and YK-2-69 group relative to DMSO group.

REVIEWERS' COMMENTS

Reviewer #4 (Remarks to the Author):

The authors have satisfactorily addressed my comments.